# Attention Illuminates LLM Reasoning: The Uncovered Preplan-and-Anchor Rhythm Enables Fine-Grained Policy Optimization

Yang Li [1 2 3]   Zhichen Dong [1 2]   Yuhan Sun [1 3]   Weixun Wang [2]   Shaopan Xiong [2]   Yijia Luo [2]
Jiashun Liu [2]   Han Lu [1 2]   Jiamang Wang [2]   Wenbo Su [2]   Bo Zheng [2]   Junchi Yan [1 3]

## Abstract

The reasoning patterns of large language models (LLMs) remain opaque, and Reinforcement learning (RL) typically assigns uniform credit across an entire generation, blurring the distinction between pivotal and routine steps. This work treats attention as a natural substrate for interpreting LLM reasoning and a window for aligning optimization with its internal dynamics. We first distinguish attention heads between locally and globally focused information processing and reveal that locally focused heads produce a sawtooth pattern near the diagonal indicating phrasal chunks, while globally focused heads expose tokens that exert broad downstream influence over future tokens. We quantify these with two metrics measuring the extent of backward attention within a clipped window and the average attention a token receives from subsequent tokens, respectively. Taken together, these signals indicate a recurring preplan-and-anchor regularity, where the model first performs a long-range contextual reference to generate an introductory token, which is immediately followed by or coincides with a semantic anchor token that organizes subsequent reasoning. Leveraging these insights, we introduce three novel RL strategies that dynamically perform targeted credit assignment to critical nodes (preplan tokens, anchor tokens, and their temporal coupling) and show consistent performance gains across various reasoning tasks.

## 1 Introduction

Large Language Models (LLMs) have achieved remarkable success on complex reasoning, propelled by training algorithms that elicit explicit step-by-step thought. A prominent paradigm is Reinforcement Learning with Verifiable Rewards (RLVR) (Lambert et al., 2024a), where a model's outputs are optimized with RL objectives guided by automated correctness checks. This setup encourages models to produce intermediate reasoning tokens before issuing a final answer, giving rise to Large Reasoning Models (LRMs) that excel on challenging problems in mathematics (Hendrycks et al., 2021; Cobbe et al., 2021b; Shao et al., 2024; Yang et al., 2024), coding (Chen et al., 2021; Jimenez et al., 2023; Hui et al., 2024; Wang et al., 2025d), and agentic decision making (Liu et al., 2023; Yao et al., 2024).

Despite impressive performance, LLMs remain largely opaque: there still lacks a clear understanding of how LLMs structure their reasoning process internally, when and how they retrieve and bind information. Consequently, optimization strategies often treat an entire generation as a uniform target. In popular RL settings, sequence-level rewards are typically distributed across all tokens (Shao et al., 2024; Yu et al., 2025), blurring the distinction between pivotal moments that shape downstream reasoning and routine steps that merely elaborate on local structure. This mismatch between how models appear to reason and how we optimize them limits data efficiency, interpretability, and the reliability of performance gains on challenging tasks. This paper seeks to bridge this gap by first showing a recurring attention-based reasoning rhythm through the lens of attention dynamics and subsequently introducing fine-grained RL strategies that enhance credit assignment to the tokens where the model appears to plan and anchor its reasoning.

Our analysis takes two complementary views of attention: a local (backward) perspective that measures how strongly a token depends on nearby versus distant context, and a global (forward) perspective that measures a token's downstream influence on subsequent tokens. Specifically, we classify attention heads as either locally or globally focused based on their average attention span, which we define as the attention-weighted mean positional distance of their at-

---

[1]Shanghai Jiao Tong University [2]Alibaba Group [3]Shanghai Innovation Institute. Correspondence to: Junchi Yan <yanjunchi@sjtu.edu.cn>, Wenbo Su <vincent.swb@alibaba-inc.com>.

*Proceedings of the $43^{rd}$ International Conference on Machine Learning*, Seoul, South Korea. PMLR 306, 2026. Copyright 2026 by the author(s).

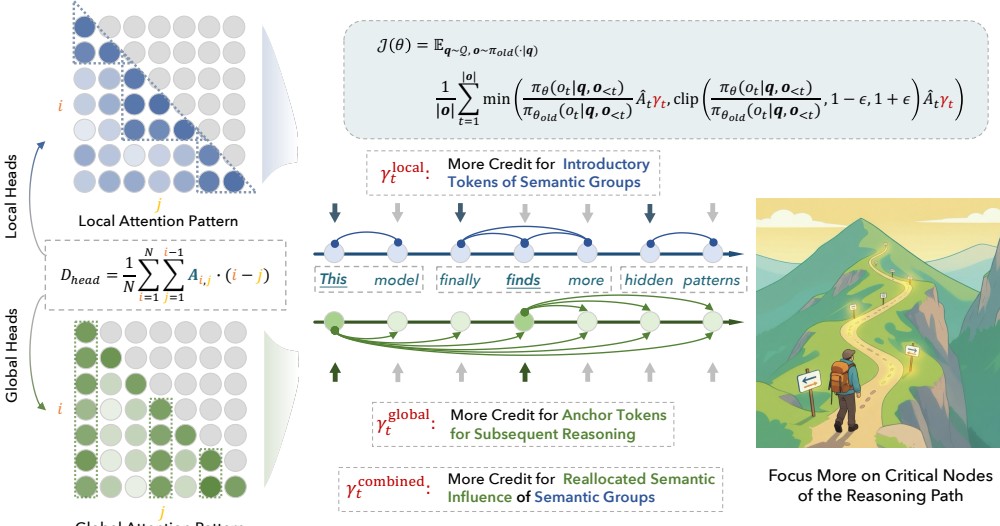

*Figure 1.* **Attention dynamics uncover reasoning patterns:** Local heads exhibit a near-diagonal, sawtooth pattern that forms phrasal chunks, while global heads highlight tokens with strong downstream influence. Their coupling reveals a preplan-and-anchor rhythm: at chunk onsets, preplan tokens set up the next reasoning step, followed (or coinciding) by anchor tokens that are persistently revisited by future tokens to guide downstream reasoning. **Reasoning rhythm enhances RL:** Leveraging detected preplan and anchor tokens, our RL method amplifies their token-level credit accordingly, focusing learning on the critical nodes that drive more effective reasoning.

tention connections. Aggregating patterns within each class reveals two robust regularities as shown in Fig. 1. First, locally focused heads exhibit a sawtooth motif near the diagonal that tracks phrasal or semantic chunks. Within a chunk (e.g., habitual expressions), attention stays highly localized, while the onset of a new chunk pulls attention back to earlier context. Second, globally focused heads highlight a small set of anchor tokens that exert broad downstream influence. These tokens are revisited by many subsequent positions and act as semantic pivots that steer the unfolding reasoning.

We distill these regularities into two model-internal metrics computed from the attention map of an auxiliary forward pass. The **Windowed Average Attention Distance (WAAD, Def. 4.1)** measures how far a token looks back within a clipped window, emphasizing whether the model must reach beyond immediate neighbors to resolve ambiguity. Visualizing the WAAD sequence reveals a clear alternation between peaks and valleys, aligned with chunk boundaries. The **Future Attention Influence (FAI, Def. 4.2)** quantifies a token's global importance by measuring the average attention it receives from later tokens. Visualizing the FAI sequence reveals differences and fluctuations in token influence, where high-influence tokens essentially correspond to key logical waypoints, such as pivotal definitions, intermediate results, or decision points. **The joint dynamics of the metrics** reveal a consistent two-beat preplan-and-anchor pattern. As the model approaches a semantic boundary, WAAD spikes as it consults longer-range context to draft an introductory token. This spike is often followed by, or coincides with, the emission of an anchor token with high FAI that organizes downstream inference. Additionally, WAAD

spikes typically maintain high token entropies. The model reaches further back when nearby cues are insufficient to disambiguate the next step (e.g., when the preceding words do not naturally license the next token). Conversely, locally dominated tokens tend to have lower entropy, reflecting stereotyped continuations.

Building on the insights, we propose fine-grained policy optimization strategies that align credit assignment with the model's attention-derived reasoning patterns. Rather than applying uniform advantages across a trajectory, we offer a process-aware alternative by leveraging internal, attention-based signals to differentially reinforce the steps the model itself treats as structurally decisive, as shown in Fig. 1: (1) emphasizing introductory tokens at chunk onsets (WAAD peaks) to establish the local scaffold for subsequent steps; (2) amplifying anchor tokens (high FAI) to articulate and preserve core semantic commitments that organize downstream inference, accelerating the propagation of verifiable signals to key decision points; and (3) when an anchor is locally dominated, reallocate part of its credit to the associated introductory token, promoting coherent chunk-level credit assignment rather than overfitting a single position.

**Our contributions include:** (1) We introduce attention dynamics as a principled lens for uncovering model-internal reasoning structure in LLMs. We formalize the patterns through two novel metrics and analyze their joint behavior to characterize local phrasal processing and global contextual anchoring. (2) We propose three novel RL methods that reweights token-level advantages to align with the models' attention-derived reasoning rhythm, emphasizing introductory and anchor tokens and their temporal coupling. (3)

Stable empirical gains across various reasoning benchmarks demonstrate our effectiveness and efficiency while maintaining compatibility with existing RLVR workflows.

## 2 Related Work

**RL for LLMs.** RL is central to LLM post-training, popularized by RLHF for instruction following and preference alignment (Christiano et al., 2017; Ziegler et al., 2019; Ouyang et al., 2022; Bai et al., 2022; Achiam et al., 2023). Two strands dominate: online policy-gradient methods with on-policy rollouts (Schulman et al., 2017; Shao et al., 2024; Williams, 1992) and offline preference optimization without on-policy sampling (Rafailov et al., 2023; Meng et al., 2024; Ethayarajh et al., 2024). Outcome-based **RL with verifiable rewards (RLVR)** (Shao et al., 2024; Lambert et al., 2024b) has driven recent advances through deterministic rule-based rewards (Xin et al., 2025; Wang et al., 2024), with large-scale systems demonstrating that correctness-guided RL elicits extended reasoning (Lambert et al., 2024a; Jaech et al., 2024; Guo et al., 2025; Team et al., 2025; Yang et al., 2025a). A growing ecosystem of studies focuses on data aggregation, policy updates, scalable training infrastructure, and controllable exploration (Yu et al., 2025; Yue et al., 2025; Zheng et al., 2025; Lu et al., 2025; Long et al., 2025).

**Analyses of LLM Reasoning.** A complementary literature analyzes how LLMs represent and execute reasoning. Step-level edits suggest surface CoT redundancy (Wang et al., 2022; Chen et al., 2024; Han et al., 2025). White-box studies identify components that propagate information, such as iteration and receiver heads (Cabannes et al., 2024; Ren et al., 2024; Bertolazzi et al., 2025), critical subsets of attention heads (Mohebbi et al., 2023; Zheng et al., 2024), steerable planning/backtracking directions (Turner et al., 2023; Venhoff et al., 2025; Zhou et al., 2024; Dong et al., 2025), and deduction circuits (Ameisen et al., 2025; Dong et al., 2024; Lindsey et al., 2025). Additional phenomena include attention sinks on initial tokens (Xiao et al., 2023) and anchor sentences that guide downstream inference (Bogdan et al., 2025). These accounts, however, are often descriptive and rarely yield process-aware training recipes.

Within RLVR, analyses indicate that gains are driven by the emergence and macro-structure of explicit reasoning rather than answer correctness and the micro-content (Gandhi et al., 2025; Li et al., 2025a). Other work targets critical tokens as decision points for exploration (Vassoyan et al., 2025; Lin et al., 2024) and emphasizes high-entropy forking tokens that govern divergent paths (Wang et al., 2025b;a; Cheng et al., 2025; Cui et al., 2025). We differ by deriving model-internal signals from attention dynamics. We identify a recurring preplan-and-anchor rhythm, link key metrics to model-internal signals, and use these signals to perform targeted credit assignment within standard RL frameworks.

## 3 Preliminaries and Notations

Given the sampled question $\mathbf{q} \sim Q$, a decoder-only LLM policy $\pi_\theta$ autoregressively generates an output sequence $\mathbf{o}$, with $\mathbf{o}_t$ as the $t$-th token in $\mathbf{o}$: $p_\theta(\mathbf{o} \mid \mathbf{q}) = \prod_{t=1}^{T} p_\theta(o_t \mid \mathbf{q}, \mathbf{o}_{<t})$. RL optimizes the LLM policy to maximize the cumulative rewards $r$ received from the verifier. Proximal Policy Optimization (PPO) (Schulman et al., 2017) is a standard approach for efficient fine-tuning, which updates $\pi_\theta$ using data from a frozen old policy $\pi_{\text{old}}$ and a clipped surrogate objective:

$$
\begin{aligned}
\mathcal{J}(\theta) = \mathbb{E}_{\mathbf{q} \sim Q, \mathbf{o} \sim \pi_\theta(\cdot|\mathbf{q})} & \frac{1}{|\mathbf{o}|} \sum_{t=1}^{|\mathbf{o}|} \min \left( \frac{\pi_\theta(o_t|\mathbf{q}, \mathbf{o}_{<t})}{\pi_{\theta_{\text{old}}}(o_t|\mathbf{q}, \mathbf{o}_{<t})} A_t, \right. \\
& \left. \text{clip} \left( \frac{\pi_\theta(o_t|\mathbf{q}, \mathbf{o}_{<t})}{\pi_{\theta_{\text{old}}}(o_t|\mathbf{q}, \mathbf{o}_{<t})}, 1-\epsilon, 1+\epsilon \right) A_t \right)
\end{aligned}
\tag{1}
$$

where $A_t$ is the advantage at step $t$, typically estimated via Generalized Advantage Estimation (GAE) (Schulman et al., 2015), and $\epsilon$ is a clipping hyperparameter for stabilizing updates. Group Relative Policy Optimization (GRPO) (Shao et al., 2024) replaces the critic with group-based advantage estimation. For a prompt $\mathbf{q}$ with $G$ responses and rewards $\{r_i\}_{i=1}^{G}$, it computes $\hat{A}_{i,t} = \frac{r_i - \text{mean}(\{r_i\}_{i=1}^{G})}{\text{std}(\{r_i\}_{i=1}^{G})}$, enhancing gradient signal reliability under sparse rewards. GRPO further augments Eq. (1) with a KL penalty to stabilize training.

## 4 Dissecting Attention Dynamics to Expose the Reasoning Rhythm

This section dissects attention dynamics to uncover a recurring model-internal reasoning rhythm and grounds each claim with **inline empirical validations**. More details are in Appendix C.1.

### 4.1 Head Grouping and the Local/Global Patterns

To uncover characteristic reasoning patterns, we analyze attention maps from two complementary directions, i.e., forward (the reach of a token's downstream influence into distant later steps) and backward (the extent to which a token's generation is dominated by the immediately preceding context). Together, these perspectives correspond to local and global views of the reasoning structure.

We analyze attention dynamics using Qwen3-4B-Base (Yang et al., 2025a) on math prompts from the GSM8K (Cobbe et al., 2021b) with temperature $T{=}0.7$. Using the non-SFT model avoids confounds from supervised instructions and provides a clean starting point for zero-RL training. For each prompt, we obtain attention maps via a single forward pass over the concatenated prompt-response sequence; unless otherwise noted, all token-level metrics are computed on rows corresponding to response tokens.

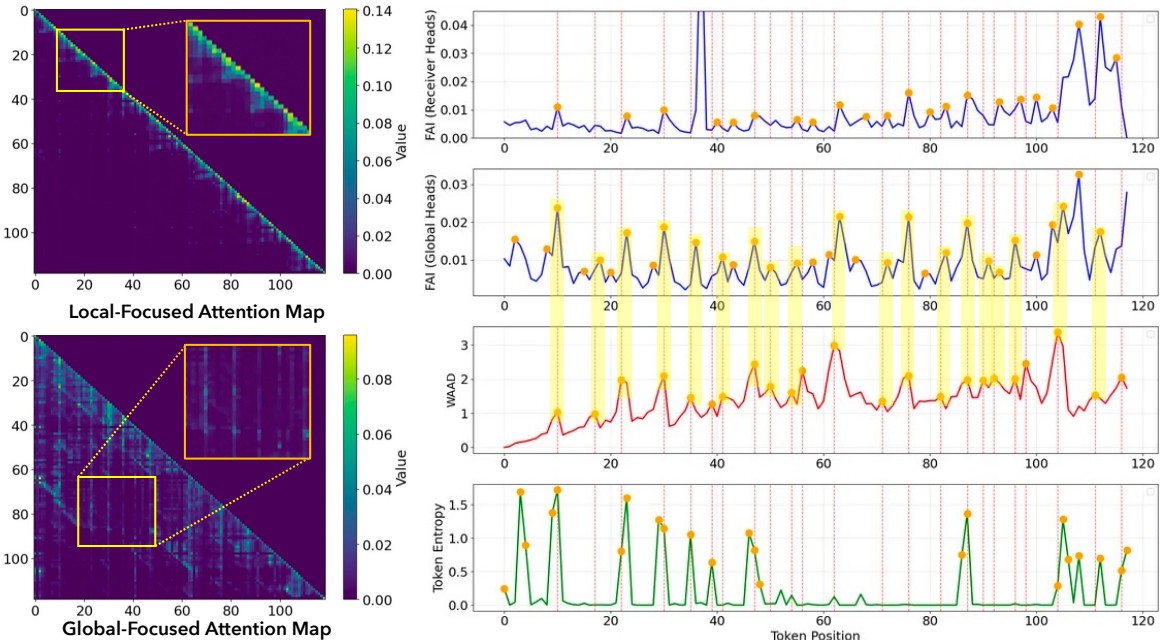

*Figure 2.* Visualization and quantification of attention patterns on a math reasoning case. Left: a local-focused attention map highlights near-diagonal sawtooth patterns, and a global-focused map highlights vertical stripes indicative of broader attention. Right: token-level metrics (FAI for receiver and global heads, WAAD, and token entropy) highlight aligned peaks that mark salient positions.

Let $\mathbf{A}^{(l,h)} \in \mathbb{R}^{N \times N}$ denote the causal attention map for layer $l$ and head $h$ over the full sequence of length $N$, with $\mathbf{A}_{t,s}^{(l,h)}$ the attention from position $t$ to $s \leq t$. Since the attention is causal and row-stochastic ($\sum_{s=1}^{t} \mathbf{A}_{t,s}^{(l,h)} = 1$ for each $t$), we can interpret each row as a probability distribution over past positions. To characterize the effective receptive span of a head, we define its attention-weighted mean backward distance over the generated response positions $\mathcal{R}$ as:

$$d^{(l,h)} = \frac{1}{|\mathcal{R}|} \sum_{t \in \mathcal{R}} \sum_{s=1}^{t} \mathbf{A}_{t,s}^{(l,h)} (t-s), \qquad (2)$$

Since the inner sum is a convex combination of distances $(t - s)$, $d^{(l,h)}$ is precisely the average distance that head $(l, h)$ looks back when generating tokens in the response. Consequently, heads with smaller $d^{(l,h)}$ indicate a strong focus toward recent context (i.e., attention concentrated near the diagonal), reflecting local focus, whereas large values signify frequent long-range dependencies (i.e., attention reaching far into the past), characteristic of global focus. We sort all heads by $d^{(l,h)}$ and designate the lowest and highest quantiles (e.g., bottom/top 30%) as the local-focused set $\mathcal{H}_{\mathrm{loc}}$ and global-focused set $\mathcal{H}_{\mathrm{glob}}$, respectively. We then aggregate attention within each group by averaging and obtain $\bar{\mathbf{A}}^{\mathrm{loc}}$ and $\bar{\mathbf{A}}^{\mathrm{glob}}$. Fig. 2 visualizes attention maps and reveals two complementary patterns.

**Local Attention Pattern: Near-Diagonal Sawtooth Indicating Local Phrasal Chunks.** The local-focused aggregate map $\bar{\mathbf{A}}^{\mathrm{loc}}$ exhibits a characteristic sawtooth along the di-

agonal that tracks phrasal or semantic chunks. Within a semantic or phrasal chunk (e.g., "by the way"), attention remains tightly local, while at the onset of a new chunk, attention abruptly reaches further back, followed by locally dominated tokens.

To operationalize this local structure, we introduce a windowed distance to measure how far a token looks back within a clipped window, emphasizing whether the model must reach beyond immediate neighbors to resolve ambiguity. We introduce a time window to downweight absolute positional artifacts since tokens of different positions handle different preceding attention ranges.

**Definition 4.1** (Windowed Average Attention Distance (WAAD)). For a response position $t \in \mathcal{R}$, with a small clipping window $W$,

$$\mathrm{WAAD}_t = \sum_{s=1}^{t} \bar{\mathbf{A}}_{t,s}^{\mathrm{loc}} \min\left(t-s, W\right). \qquad (3)$$

Low WAAD values indicate tight local continuation within a chunk (valleys), while spikes represent long-range consultation at chunk boundaries (peaks). The WAAD sequence in Fig. 2 reveals clear peak-valley alternation aligned with chunk boundaries: a chunk onset typically exhibits a peak (it must retrieve earlier context to form semantics), followed by a sharp drop as subsequent tokens rely on immediate neighbors. Mechanistically, local heads enact habit-driven micro-syntax, where once a chunk is initiated, subsequent tokens are locally licensed with low uncertainty, reflect-

*Table 1.* Jaccard similarity of rollouts for top- and bottom-$k$ ranked by FAI. $\Pr(\text{Top} < \text{Bottom})$ denotes the percentage of trials where top-$k$ similarity is lower.

|  | Top-$k$ | Bottom-$k$ |
| --- | --- | --- |
| Mean Jaccard similarity | 0.534 | 0.631 |
| $\Pr(\text{Top} < \text{Bottom})$ | | 87.14% |

ing collocational continuations internalized during pretraining. Hesitation concentrates at chunk boundaries, where the model consults longer-range context.

**Global Attention Pattern: Sparse Anchors with Broad Influence to Downstream Tokens.** The global-focused aggregate map $\bar{\mathbf{A}}^{\text{glob}}$ disproportionately highlights a sparse set of tokens that receive sustained attention from many future positions. These tokens function as semantic anchors that are repeatedly revisited and steer unfolding reasoning to a stable frame. We quantify a token's downstream importance by averaging the attention it receives from future positions within a controlled horizon.

**Definition 4.2** (Future Attention Influence (FAI))**.** For a token $s$, let $H_{\text{lo}}$ and $H_{\text{hi}}$ be lower/upper horizons for influence calculation. With the index set defined as $\mathcal{T}(s) = \{ t \mid t \in \mathcal{R}, \ s + H_{\text{lo}} \leq t \leq \min(N, \ s + H_{\text{hi}}) \}$,

$$\text{FAI}_s = \frac{1}{|\mathcal{T}(s)|} \sum_{t \in \mathcal{T}(s)} \bar{\mathbf{A}}^{\text{glob}}_{t,s}. \tag{4}$$

High FAI surfaces tokens with broad downstream reach (anchor tokens). In Fig. 2, visualizing FAI over a response reveals differences and fluctuations in token influence, where high-influence tokens essentially correspond to key logical waypoints (e.g., pivotal definitions, intermediate results, or decision points). We also observe a growing density of high-FAI tokens as the trajectory approaches the final answer, consistent with the need to consolidate prior reasoning.

**Do Perturbations at High-FAI Tokens More Effectively Alter Subsequent Reasoning?** At a chosen position $i^*$, we take the predictive distribution $p_\theta(\cdot \mid \mathbf{o}_{<i^*})$, select the top-$k$ candidates $\mathcal{V}_k(i^*)$ (i.e., highest-logit tokens), and for each $v \in \mathcal{V}_k(i^*)$ we force $o_{i^*} = v$ and roll out continuations under greedy decoding. We then compare each counterfactual trajectory $\tilde{\mathbf{o}}^v_{>i^*}$ with the original $\mathbf{o}_{>i^*}$ using Jaccard overlap over content words, where $\mathcal{S}(\cdot)$ denotes the content token set of a reasoning trajectory:

$$\text{Jaccard}\left(\mathbf{o}_{>i^*}, \tilde{\mathbf{o}}^v_{>i^*}\right) = \frac{\left|\mathcal{S}(\mathbf{o}_{>i^*}) \cap \mathcal{S}(\tilde{\mathbf{o}}^v_{>i^*})\right|}{\left|\mathcal{S}(\mathbf{o}_{>i^*}) \cup \mathcal{S}(\tilde{\mathbf{o}}^v_{>i^*})\right|}. \tag{5}$$

We conduct this analysis on 70 randomly sampled math problems. We conduct rollout simulations on high-FAI positions and low-FAI positions for comparison. As shown in Table 1, across problems and rollout horizons, perturbations at high-FAI positions yield substantially lower Jaccard

similarity, with 87.14% of trials showing greater deviation than at low-FAI positions. This indicates that FAI-identified anchors serve as useful proxies for influential positions: changing them reshapes the global plan and cadence of reasoning, whereas interventions at locally dominated positions primarily affect surface form without redirecting the overall trajectory. Examples are shown in Appendix D.10.

### 4.2 Joint Dynamics: A Preplan-and-Anchor Pattern

Single-metric evaluations of reasoning often provide an incomplete picture. In practice, key internal signals are closely coupled. This section explores the joint dynamics of internal metrics while also connecting to readily observable indicators. A multidimensional analysis of these signals uncovers three robust empirical couplings, converging on a recurring preplan-and-anchor regularity that reflects coordinated local-global attention dynamics.

**Coupling Pattern 1: WAAD Peaks Maintain Higher Token Entropies.** Token-level entropy $H_t = -\sum_v p_\theta(v \mid \mathbf{q}, \mathbf{o}_{<t}) \log p_\theta(v \mid \cdot)$ reflects predictive uncertainty and drives reasoning exploration (Wang et al., 2025b). As shown in Fig. 2, spikes in $\text{WAAD}_t$ correspond to tokens with higher $H_t$. When immediate local cues dominate, next-token prediction is confident (low $H_t$) and attention remains near-diagonal (low $\text{WAAD}_t$), which is consistent with habit-driven micro-syntax learned in pretraining (e.g., after "by the", "way" follows naturally). In contrast, at semantic junctures where the local context underdetermines the next step, the model exhibits higher uncertainty (high $H_t$) and reaches further back to retrieve disambiguating information (high $\text{WAAD}_t$). In this sense, high-entropy tokens and WAAD peaks can be two sides of the same coin: local ambiguity triggers long-range consultation. Conversely, locally dominated tokens tend to have lower entropy, reflecting stereotyped continuations led by local context.

**Coupling Pattern 2: Receiver Heads and Global-Focused Heads Surface Shared Anchors.** Bogdan et al. (2025) identify receiver heads, i.e., attention heads whose inbound attention distribution is highly selective and thus diagnostic of tokens that attract strong attention from future positions (e.g., via column-wise kurtosis). We compare those heads with our global-focused set $\mathcal{H}_{\text{glob}}$, and find that the token-level downstream influence computed from either set is highly concordant. Aggregating attention over receiver heads or over $\mathcal{H}_{\text{glob}}$ yields strongly correlated FAI profiles across tokens, as shown in Fig. 2, indicating that span-based and kurtosis-based filtering arrive at the same conclusions and mutually corroborate one another.

**Coupling Pattern 3: FAI Peaks Follow or Coincide with WAAD Peaks.** As highlighted in Fig. 2, high-FAI tokens typically occur at or just after WAAD peaks, reflecting a two-beat process: (i) *Preplan*: As the model approaches a se-

*Table 2.* Quantitative linkages between metrics. Random baselines are expectation references based on the probability under completely random conditions.

| Metric Correlation | Random | Observed | Lift |
|---|---|---|---|
| Average Entropy of the WAAD Peaks | 0.2386 | 0.3608 | +51.97% |
| FAI Peak of Receiver & Global Heads Co-occurrence | 22.41% | 60.84% | +171.49% |
| FAI Peak Follows/Coincides with WAAD Peak | 36.87% | 52.53% | +42.47% |

mantic boundary, a WAAD peak signals long-range context retrieval to generate an introductory token that prepares the forthcoming concept or step. (ii) *Anchor*: At the same position or immediately later, the model emits a token with high FAI, repeatedly attended by future positions to guide and stabilize subsequent reasoning. Notably, key logical content can be preceded by an introductory token that frames the upcoming idea, possibly to maintain fluency or adhere to conventional phrases. From a local perspective, such anchor tokens can be dominated by immediately preceding tokens, leaving little room for exploration at the anchor position itself. This motivates joint consideration of both the anchor and its introductory token during optimization.

**Quantitative Analysis.** We sample 70 questions and analyze linkages between metrics for quantification. For each coupling, we compute the observed statistic (e.g., average entropy at WAAD peaks, co-occurrence rates of FAI and receiver-head peaks, alignment between WAAD and FAI peaks) and compare it against the expectation under random situations in which the positions of the first metric's peaks are shuffled. Table 2 shows that all couplings exhibit substantial lifts over random chance (ranging from +42.47% to +171.49%), validating these joint dynamics.

## 5 Fine-Grained Policy Optimization Driven by Attention Signals

We align token-level credit with the model's attention-derived reasoning rhythm by rescaling the per-token advantage $A_t$ in on-policy RL using data-dependent weights derived from attention dynamics:

$$\tilde{A}_t = A_t \gamma_t \qquad (6)$$

where $\gamma_t$ is the scaling coefficient. Unlike continuous reweighting, our variants amplify the advantages of tokens identified by the model's internal reasoning patterns, thereby improving the efficiency of RL. We instantiate $\gamma_t$ with three strategies that correspond to the local preplan signal, the global anchor signal, and their joint coupling. The model typically requires thousands of forward passes to generate a response, whereas we obtain attention maps with just a single forward pass over the generated response (together with the prompt), introducing negligible computational overhead. **Implementation details** of calculating full attention during RL are given in Appendix A.

**(1) Local-chunk credit: select preplan tokens via distance drops.** For tokens within each response, we detect

preplan tokens by the WAAD variations between consecutive positions and then select the top $q$ quantile tokens:

$$\Delta_t = |\text{WAAD}_t - \text{WAAD}_{t+1}| \quad \mathcal{T}_{\text{loc}} = \text{TopQ}(\Delta_t, q), \qquad (7)$$

Here, large $\Delta_t$ identifies the boundary tokens at a peak-valley transition of phrasal chunks. Since these tokens initiate local scaffolds and guide subsequent reasoning, emphasizing these points strengthens the locus of planning and encourages the policy to resolve long-range dependencies before committing. We amplify advantages of tokens within $\mathcal{T}_{\text{loc}}$ by configuring the scaling coefficient:

$$\gamma_t = 1 + (\gamma_{\text{amp}} - 1) \, \mathbf{1}\{t \in \mathcal{T}_{\text{loc}}\}, \qquad (8)$$

where $\gamma_{\text{amp}} = 1.5$ denotes an amplification factor.

**(2) Global-anchor credit: select tokens with high future influence.** We score tokens by FAI and then select the top $q$ quantile tokens accordingly:

$$\mathcal{T}_{\text{glob}} = \text{TopQ}(\text{FAI}, q{=}0.4), \qquad (9)$$

By amplifying anchors, it learns to articulate and preserve core semantic commitments that organize downstream inference. Because these tokens largely determine subsequent reasoning, prioritizing reinforcement (and, when applicable, penalties) at these positions propagates the verifiable signal to the key points more quickly, enabling more targeted optimization. Thus, We amplify advantages of tokens within $\mathcal{T}_{\text{glob}}$ by configuring the scaling coefficient:

$$\gamma_t = 1 + (\gamma_{\text{amp}} - 1) \, \mathbf{1}\{t \in \mathcal{T}_{\text{glob}}\}, \qquad (10)$$

where $\gamma_{\text{amp}} = 1.5$ denotes an amplification factor.

**(3) Coupled rhythm credit: combine preplans and anchors, and back-allocate credit from anchors to their local precursors.** A locally dominated anchor has limited freedom to optimize, so we back-allocate credit of high-FAI tokens across the local semantic chunk, promoting coherent chunk-level scaffolding rather than overfitting to a single position. We couple WAAD with FAI signal to reflect the two-beat rhythm and to share credit between anchors and their preceding preplans. The key idea is: identify high-influence anchors, test whether they are locally dominated, and, if so, diffuse a fraction of their credit backward to the preplan token that prepares the anchor.

With the anchor candidate set $\mathcal{T}_{\text{glob}}$, the WAAD variations $\Delta_t$, and given thresholds $\tau_{\text{waad}}$ and $\tau_\Delta$, we say an anchor $t$ is locally dominated by its immediate past $k$ tokens if

$$\text{WAAD}_t \leq \tau_{\text{waad}} \text{ and } \max_{u \in \{t-k, \dots, t-1\}} \Delta_u \geq \tau_\Delta. \qquad (11)$$

We denote $t \in \mathcal{D}$ when $t$ is a high-influence token with low WAAD, locally licensed preceded by a recent WAAD peak.

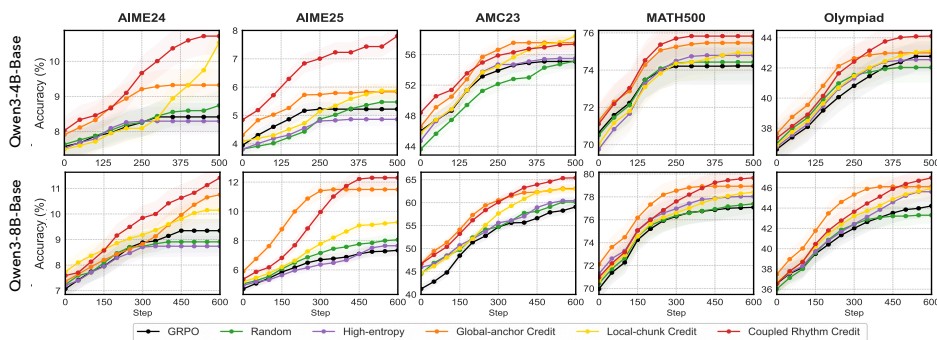

*Figure 3.* RL Performance curves of Qwen3-4B-Base and Qwen3-8B-Base models for math reasoning.

We redistribute a fraction $\alpha \in [0, 1]$ of the amplification bonus $\gamma_{\mathrm{amp}}-1$ from each locally dominated anchor to its associated introductory token $\mathcal{I}(\mathcal{D}) = \{\mathrm{intro}(s) : s \in \mathcal{D}\}$:

$$
\gamma_t = 1 + (\gamma_{\mathrm{amp}}-1)\,\mathbf{1}\{t \in \mathcal{T}_{\mathrm{glob}} \setminus \mathcal{D}\} \\
+ (1-\alpha)(\gamma_{\mathrm{amp}}-1)\,\mathbf{1}\{t \in \mathcal{D}\} + \alpha(\gamma_{\mathrm{amp}}-1)\,\mathbf{1}\{t \in \mathcal{I}(\mathcal{D})\}, \quad (12)
$$

where $\gamma_{\mathrm{amp}} = 1.5$ denotes an amplification factor.

### 5.1 Experimental Evaluation

#### 5.1.1 EXPERIMENTAL SETTINGS

**Benchmarks.** We test on relatively simple puzzles and question-answering (QA) benchmarks and challenging mathematical reasoning datasets. (i) the *Countdown* puzzle (Pan et al., 2025), where the goal is to combine four given numbers using arithmetic operations to reach a target value; (ii) the *CrossThink-QA* dataset (Akter et al., 2025), which aggregates multi-domain QA pairs from heterogeneous sources; and (iii) five standard math reasoning benchmarks: AIME24, AIME25, AMC, MATH500 (Hendrycks et al., 2021), and OlympiadBench (He et al., 2024).

**Baselines.** We introduce a novel modulation of token-level advantages for RL. To isolate the impact of this design, we implement our method on top of GRPO to directly assess the improvement. In addition, we consider alternative token-level advantage enhancement strategies for comparison: (1) *Random*: randomly select tokens for advantage amplification; (2) *Entropy*: amplify advantages for tokens with high predictive entropy to encourage exploration.

**Backbone Models and Implementation Details.** Experiments are conducted using the Qwen3-4B-Base and Qwen3-8B-Base models. We use a maximum context length of 1024 for simple puzzles and QA, and 1024 or 8192 for math reasoning. The shorter context length is adopted as it yields a cleaner reasoning process that aligns more closely with the analytical environment and is less confounded by long-range dependencies that can dilute the effect of attention-based strategies. The curves are smoothed through the exponential moving averages of the peak performance, emphasizing sustained improvements.

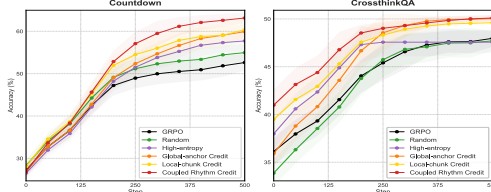

*Figure 4.* Performance curves of Qwen3-4B-Base RL processes on Countdown puzzle and QA.

*Table 3.* Results of the logical puzzle and question answering tasks. Bold denotes the best results per dataset.

| Method | Countdown | CrossThinkQA |
|---|---|---|
| GRPO | 52.6 | 48.0 |
| + random credit | 55.0 | 47.8 |
| + high-entropy credit | 57.7 | 48.0 |
| + local-chunk credit | 59.9 +7.3 | 50.0 +2.0 |
| + global-anchor credit | 60.4 +7.7 | 49.6 +1.6 |
| + coupled rhythm credit | **63.1** +10.5 | **50.1** +2.1 |

**Training details.** We use a training batch size of 512 and micro-batch size of 32, yielding 16 gradient steps per batch, with learning rate $1 \times 10^{-6}$. The loss excludes both KL and entropy regularization. Decoding temperature during training is $T=1.0$. Our rhythm-aligned variants replace the per-token advantage $A_t$ by $\tilde{A}_t = \gamma_t A_t$, where $\gamma_t$ is computed from attention maps captured in a single forward pass. Unless noted, we set the WAAD window $W = 10$, the FAI horizon $H \in [10, 100]$, anchor selection by top-quantile ($q=40\%$), neighborhood size $k \in \{1, 2, 3\}$ for back-allocation. All shaping signals are detached from gradients and applied to the signed per-token advantages, covering both positive and negative advantages. The 4B models are trained on 8 GPUs, and the 8B models are trained on 16 GPUs, running for 500 and 600 steps, respectively. More experimental details for RL experiments can be found in Appendix C.2.

#### 5.1.2 RESULTS AND ANALYSES

**Results on Simple Logical Puzzles and QA.** Table 3 reports the final accuracy on the Countdown and CrossThink-QA benchmarks. On Countdown, our *coupled rhythm credit* strategy achieves 63.1%, substantially outperforming GRPO (52.6%). Notably, both *local-chunk* and *global-anchor*

*Table 4.* Results of math reasoning. AIME 24&25: avg@16; others: pass@1.

| Method | AIME24 | AIME25 | AMC23 | MATH | Olympiad | Avg. |
|---|---|---|---|---|---|---|
| *Qwen3-4B-Base with 1K Length* | | | | | | |
| GRPO | 8.4 | 5.2 | 55.1 | 74.2 | 42.8 | 37.1 |
| + random credit | 8.7 | 5.5 | 55.2 | 74.4 | 42.0 | 37.1 |
| + high-entropy credit | 8.3 | 4.9 | 55.5 | 74.8 | 42.5 | 37.2 |
| + global-anchor credit | 9.3 +0.9 | 5.8 +0.6 | 57.6 +2.5 | 75.5 +1.2 | 43.0 +0.2 | 38.2 +1.1 |
| + local-chunk credit | 10.5 +2.1 | 5.9 +0.7 | 58.4 +3.3 | 74.9 +0.7 | 43.1 +0.4 | 38.6 +1.5 |
| + coupled rhythm credit | **10.7 +2.3** | **7.8 +2.6** | 57.4 +2.3 | **75.8 +1.6** | **44.1 +1.3** | **39.2 +2.1** |
| *Qwen3-8B-Base with 1K Length* | | | | | | |
| GRPO | 9.3 | 7.3 | 59.1 | 77.1 | 44.2 | 39.4 |
| + random credit | 8.9 | 8.1 | 60.1 | 77.4 | 43.3 | 39.6 |
| + high-entropy credit | 8.7 | 7.7 | 60.4 | 78.0 | 45.6 | 40.1 |
| + global-anchor credit | 10.8 +1.4 | 11.5 +4.2 | 63.1 +4.0 | 78.9 +1.8 | 46.1 +1.9 | 42.1 +2.7 |
| + local-chunk credit | 10.1 +0.8 | 9.3 +1.9 | 62.9 +3.8 | 78.4 +1.3 | 45.9 +1.7 | 41.3 +1.9 |
| + coupled rhythm credit | **11.4 +2.1** | **12.3 +5.0** | **65.4 +6.3** | **79.7 +2.6** | **47.0 +2.8** | **43.2 +3.8** |
| *Qwen3-4B-Base with 8K Length* | | | | | | |
| GRPO | 19.5 | 16.1 | 57.6 | 81.0 | 49.9 | 44.8 |
| + random credit | 18.0 | 16.6 | 57.1 | 82.0 | 50.0 | 44.7 |
| + high-entropy credit | 19.3 | 15.4 | 57.8 | 81.2 | 48.6 | 44.5 |
| + global-anchor credit | 21.7 +2.3 | 19.5 +3.4 | **61.1 +3.5** | 82.1 +1.2 | 51.2 +1.3 | 47.1 +2.3 |
| + local-chunk credit | **22.4 +2.9** | **20.4 +4.3** | 59.2 +1.6 | 82.5 +1.5 | 51.5 +1.6 | 47.2 +2.4 |
| + coupled rhythm credit | 22.0 +2.5 | 19.9 +3.8 | 59.3 +1.7 | **82.9 +1.9** | **52.2 +2.3** | **47.3 +2.5** |

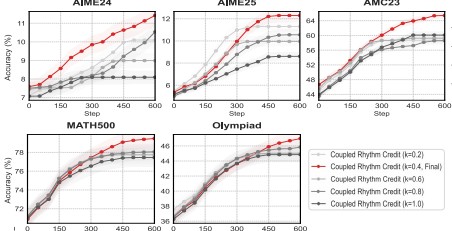

*Figure 5.* Curves of Top-k versus Bottom-k additional credit guided by the metrics.

*Figure 6.* Performance curves of coupled rhythm credit by different Top-k ratios.

credit schemes also yield consistent gains, suggesting that structured credit propagation effectively guides policy learning. In contrast, random or entropy-based token selection provides marginal or no improvement. On CrossThink-QA, where reasoning is more open-ended and less constrained by formal rules, all credit-aware variants show consistent improvements over GRPO (48.0%). The best-performing variant, *coupled rhythm credit*, reaches 50.1%, indicating that attention-based credit assignment can aid generalization in heterogeneous QA. Fig. 4 shows the training curves on Countdown and QA tasks. Our variants converge faster and reach higher plateaus than GRPO, with *coupled credit* exhibiting the earliest improvement and highest scores.

**Results on Mathematical Reasoning.** Table 4 evaluates credit assignment strategies across mathematical reasoning benchmarks and model scales. Our methods consistently outperform both the GRPO baseline and naive alternatives (e.g., random or high-entropy credit). Notably, *coupled rhythm credit* achieves the strongest gains across all settings, with the most pronounced improvements on challenging tasks for Qwen3-8B (e.g., +5.0pt on AIME25, +6.3pt on AMC23). These gains are robust across sequence lengths: with an extended 8K context, where base GRPO already benefits from longer reasoning traces, it still delivers consistent improvements (e.g., +4.3pt on AIME25). Fig. 3 further corroborates these findings, showing faster convergence and higher final performance for rhythm-aware credit strategies.

**Ablation: Top-$k$ versus Bottom-$k$.** For proposed metrics, we experiment with reallocating additional credit to tokens ranked in the bottom 40% according to these metrics. Fig. 5 and Table 5 show that reinforcing credit on such low-scoring tokens (i.e., those locally dominated or exhibiting minimal global influence) leads to degraded performance. RL train-

*Table 5.* Quantitative ablation results for Top-k vs. Bottom-k credit reallocation and different top-k ratios.

| Method | AIME24 | AIME25 | AMC23 | MATH500 | Olympiad | Avg |
|---|---|---|---|---|---|---|
| GRPO (Baseline) | 9.3 | 7.3 | 59.1 | 77.1 | 44.2 | 39.4 |
| Global (Bottom-40%) | 7.0 -2.3 | 4.8 -2.5 | 46.0 -13.1 | 71.2 -5.9 | 36.4 -7.8 | 33.1 -6.3 |
| Local (Bottom-40%) | 8.0 -1.3 | 5.1 -2.2 | 57.9 -1.2 | 74.8 -2.3 | 42.1 -2.1 | 37.6 -1.8 |
| Coupled (Bottom-40%) | 7.2 -2.1 | 4.7 -2.6 | 44.4 -14.7 | 70.3 -6.8 | 36.0 -8.2 | 32.5 -6.9 |
| Global (Top-40%) | 10.8 +1.4 | 11.5 +4.2 | 63.1 +4.0 | 78.9 +1.8 | 46.1 +1.9 | 42.1 +2.7 |
| Local (Top-40%) | 10.1 +0.8 | 9.3 +1.9 | 62.9 +3.8 | 78.4 +1.3 | 45.9 +1.7 | 41.3 +1.9 |
| Coupled (Top-40%) | **11.4 +2.1** | **12.3 +5.0** | **65.4 +6.3** | **79.7 +2.6** | **47.0 +2.8** | **43.2 +3.8** |
| Coupled (Top-20%) | 10.1 | 11.3 | 58.7 | 78.0 | 45.1 | 40.6 |
| Coupled (Top-60%) | 9.0 | 9.9 | 59.2 | 77.7 | 45.0 | 40.2 |
| Coupled (Top-80%) | 10.5 | 10.6 | 58.6 | 78.1 | 45.8 | 40.7 |
| Coupled (Top-100%) | 8.1 | 8.6 | 60.1 | 77.5 | 44.8 | 39.8 |

ing becomes ineffective, and the peak evaluation metrics show no improvement on mathematical reasoning. In contrast, preferentially assigning credit to top-$k$ tokens yields clear improvements. This contrast underscores the validity of our metric in identifying decisive positions.

**Ablation: Top-$k$ ratios.** Fig. 6 shows the performance curves of our *coupled rhythm credit* with different top-$k$ ratios, evaluated on multiple mathematical benchmarks using Qwen3-8B-Base. Table 5 shows the quantitative results. The results confirm that allocating credit exclusively to the top 40% of tokens (i.e., $k = 0.4$) yields the strongest overall performance, achieving the highest scores across all datasets and a peak average of 43.2. Both smaller ($k = 0.2$) and larger ($k \geq 0.6$) ratios lead to performance drops, indicating that reinforcing too few or too many tokens dilutes the signal for critical reasoning positions.

## 5.2 Additional Robustness and Ablation Analyses

**Robustness across model families and scales.** To examine whether our method remains effective across model families and scales, we further evaluate the coupled rhythm credit strategy on models beyond Qwen3-4B/8B. Table 6

*Table 6.* Robustness across model families and scales. Each benchmark entry reports the result of coupled rhythm credit, with the subscript denoting the absolute improvement over GRPO under the same setting. All experiments use a 1K response length.

| Category | Model | AMC23 | MATH500 | Olympiad | Minerva | GSM8K | Avg. | Rel. Gain |
|---|---|---|---|---|---|---|---|---|
| Other families | Llama3.2-3B | $4.8_{+1.0}$ | $6.3_{+0.9}$ | $2.9_{+0.7}$ | $3.3_{+0.1}$ | $11.0_{+1.5}$ | $5.7_{+0.9}$ | $+18.8\%$ |
| | Llama3.1-8B | $5.2_{+2.0}$ | $7.7_{-0.4}$ | $3.9_{+0.7}$ | $6.0_{+0.7}$ | $20.8_{+2.1}$ | $8.7_{+1.0}$ | $+13.0\%$ |
| | DeepSeek-R1-Distill-Llama-8B | $45.6_{+3.2}$ | $64.2_{+1.4}$ | $31.9_{+3.0}$ | $25.8_{+1.5}$ | $79.2_{+1.2}$ | $49.34_{+2.06}$ | $+4.36\%$ |
| Other scales | Qwen3-0.6B | $11.2_{+2.9}$ | $37.9_{+7.4}$ | $10.1_{+2.5}$ | $12.7_{+5.4}$ | $59.7_{+10.7}$ | $26.32_{+5.78}$ | $+28.14\%$ |
| | Qwen3-1.7B | $20.7_{+4.8}$ | $55.9_{+3.8}$ | $18.5_{+0.7}$ | $24.2_{+6.1}$ | $77.8_{+0.7}$ | $39.42_{+3.22}$ | $+8.90\%$ |

*Table 7.* Additional ablations. (a) Signed advantage scaling on Qwen3-8B-Base with 1K response length. The sign columns indicate whether $\gamma_t$ is applied to positive or negative advantages. (b) WAAD/FAI hyperparameter sensitivity on Qwen3-4B-Base with 1K response length, reporting average scores over five math benchmarks.

**(a) Signed advantage scaling**

| Method | Scaled sign | | Benchmark | | | | | Avg. |
|---|---|---|---|---|---|---|---|---|
| | $A_t > 0$ | $A_t < 0$ | AIME24 | AIME25 | AMC23 | MATH | Olympiad | |
| GRPO | – | – | 9.3 | 7.3 | 59.1 | 77.1 | 44.2 | 39.4 |
| Positive-only | ✓ | – | 10.6 | 10.9 | 64.5 | 77.2 | 46.6 | 42.0 |
| Negative-only | – | ✓ | 9.8 | 8.4 | 62.2 | 77.9 | 45.7 | 40.8 |
| **Both-sign** | ✓ | ✓ | **11.4** | **12.3** | **65.4** | **79.7** | **47.0** | **43.2** |

**(b) WAAD/FAI hyperparameters**

| Sweep | Best / chosen Avg. | Avg. range |
|---|---|---|
| WAAD $W$ | **38.6** at 10 | 37.0–38.6 |
| FAI $H_{\text{lo}}$ | **38.2** at 10 | 34.8–38.2 |
| FAI $H_{\text{hi}}$ | **38.2** at 100 | 37.4–38.2 |
| GRPO baseline | 37.1 | – |

reports results on Llama3.2-3B, Llama3.1-8B, DeepSeek-R1-Distill-Llama-8B, and smaller Qwen3 models, with full baseline results deferred to Appendix D.12. For compactness, each entry reports coupled rhythm credit, and the subscript denotes its absolute improvement over GRPO under the same setting. Despite the difficulty of direct RL without an SFT warm start on base Llama models, our method improves the average score by $+18.8\%$ on Llama3.2-3B and $+13.0\%$ on Llama3.1-8B, and further raises the average score on the stronger DeepSeek-R1-Distill-Llama-8B from 47.28 to 49.34. The gains also persist at smaller Qwen3 scales, improving the average score by $+5.78$ on Qwen3-0.6B and $+3.22$ on Qwen3-1.7B. These results suggest that the attention-derived credit signal is not tied to a single model family or scale.

**Ablation: signed advantage scaling.** Our method rescales the signed token-level advantage $A_t$ with $\gamma_t$, so the same attention-derived signal can emphasize both rewards and penalties at rhythm-selected positions. To isolate the effect of this signed scaling, we compare three variants that apply $\gamma_t$ to positive advantages only, negative advantages only, or both signs. As shown in Table 7(a), either one-sided variant improves over GRPO, while both-sign scaling performs best across all benchmarks, raising the average score from 39.4 to 43.2. We therefore use both-sign scaling in all main experiments.

**Ablation: WAAD/FAI hyperparameters.** We ablate the metric-level hyperparameters that determine the temporal scope of WAAD and FAI, varying each parameter around its default while keeping the others fixed. Since $W$ affects local-chunk credit and $H_{\text{lo}}, H_{\text{hi}}$ affect global-anchor credit, each sweep is conducted under the corresponding single-signal variant. As summarized in Table 7(b), the chosen

values achieve the best average performance in their respective sweeps, while most nearby choices remain above GRPO. This suggests that our method is reasonably robust to metric-level hyperparameter choices. Full per-benchmark results are provided in Appendix D.13.

**Scope of attention-derived signals.** We clarify that our method does not treat attention maps as unbiased explanations or complete causal decompositions of reasoning. Instead, WAAD and FAI serve as filtered model-internal signals for locating structurally influential positions in a generated trajectory. The RL objective remains unchanged; attention only determines where the signed sequence-level advantage is reweighted. Our approach does not assess the local correctness of each intermediate step, but reallocates sequence-level feedback to tokens that more strongly shape the trajectory. Thus, the preplan-and-anchor rhythm should be viewed as an attention-derived regularity useful for credit assignment, rather than a complete theory of reasoning.

## 6 Conclusion

We show that analyzing attention dynamics provides a powerful new lens for understanding and directing LLM reasoning and post-training designs. By analyzing local and global attention, we uncover a recurring preplan-and-anchor rhythm in LLM reasoning: long-range consultation precedes anchor tokens that organize downstream inference. We formalize this with two metrics, WAAD and FAI, that identify preplan and anchor tokens. Using these signals, we design a targeted RL credit assignment that amplifies rewards on critical nodes, and delivers consistent gains on reasoning tasks. Aligning optimization with models' internal rhythm opens a new path to more transparent and effective training.

## Acknowledgment

This work was in part supported by Scientific Research Innovation Capability Support Project for Young Faculty (U40) of the Ministry of Education of China (SRICSPYF-ZY2025019), NSFC 625B2119 and Alibaba Group.

## Impact Statement

This paper contributes not only an RL algorithm, but also an analysis-driven perspective on reasoning patterns that informs how credit should be assigned. Specifically, this work asks and operationalizes a key question for a potential new perspective of RL4LLM: *can deeper access to a model's internal reasoning patterns enable more effective reinforcement of reasoning ability?* We elevate *attention* from a computation byproduct to a mechanistic trace that can reveal the model's internal reasoning rhythm, e.g., when it is deliberating versus retrieving, which tokens/steps act as decision bottlenecks, and where the trajectory branches into success or failure. This enables a new entry point for RL: optimizing policies in ways that are explicitly aligned with the model's attention-derived reasoning patterns.

This direction has further potential impacts to the field. First, it can improve sample efficiency and stability by focusing learning signals on causally important reasoning segments, thereby addressing a core bottleneck in long-horizon credit assignment. Second, it can increase transparency: the internal structure used to guide learning also provides a legible explanation of *why* updates occur, strengthening interpretability and auditability. Third, it motivates a broader paradigm for training reasoning models in which behavior analysis and optimization can be tightly coupled, opening additional opportunities for principled improvements and accelerating progress in RL4LLM.

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

# Appendix

# A  Attention Calculation in RL Framework

**The Vanilla RL Framework.** In reinforcement learning (RL) for large language models (LLMs), the core objective is to optimize a policy that maximizes expected task-specific rewards, such as correctness in mathematical reasoning, while maintaining alignment with the original language distribution. Under PPO framework, which also generalizes to GRPO and similar on-policy methods, training proceeds through an iterative loop of inference, reward evaluation, and policy update. We implement our methods on top of the ROLL framework (Wang et al., 2025c).

In our setup, two reasoning model instances are maintained: (1) `actor_infer`: deployed with vLLM for high-throughput, low-latency inference; (2) `actor_train`: implemented in Megatron for efficient large-scale training with model parallelism. At each iteration, `actor_infer` (initialized from the latest `actor_train` weights) generates a batch of responses to input prompts. These responses are evaluated by a reward model or an execution-based verifier, yielding a scalar reward per sequence. The prompt-response-reward tuples constitute the RL data. This data is then fed to `actor_train`, which computes policy gradients using the clipped objective to ensure stable updates. After each update, the updated weights of `actor_train` are synchronized to `actor_infer`, closing the loop.

**Implementation of Attention Mechanisms in RL.** Modern distributed training frameworks and inference engines for LLM like vLLM and Megatron typically employ flash attention (Dao et al., 2022) for computational efficiency, which discards full attention matrices during execution to reduce memory overhead. Consequently, the complete attention maps are not available for analysis in either `actor_infer` or `actor_train`. To address this, we introduce a dedicated auxiliary model, `actor_attn`, implemented with a standard Transformer. This model retains full attention weights during the forward pass, enabling us to compute internal metrics based on attention dynamics.

Specifically, after `actor_infer` completes generation of a response (which involves hundreds to thousands of forward steps during autoregressive decoding), we concatenate the original prompt and the generated response into a single sequence and perform one additional forward pass through `actor_attn`. During this pass, we sample attention maps from five evenly spaced layers within the middle third of the network (i.e., from layers $\lfloor L/3 \rfloor$ to $\lfloor 2L/3 \rfloor$, where $L$ is the total number of layers) to serve as representative snapshots of the model's internal attention behavior. The model typically requires thousands of forward passes for token-by-token generation, whereas we obtain attention maps with just a single additional forward pass over the generated

response (together with the prompt), introducing little additional latency with parallel computation. After each policy update of `actor_train`, its parameters are synchronized to both `actor_infer` and `actor_attn` to ensure consistency across inference, training, and attention analysis.

# B  Detailed Preliminaries

## B.1  Self-Attention in Decoder-Only LLMs

Generative modeling has shown great promise across a wide range of scenarios (Li et al., 2022; 2024b; 2025c; 2026; 2025b; Chen et al., 2026; Wang et al., 2026), with large language models (LLMs) being a prime example. Given the sampled question $\mathbf{q} \sim Q$, a decoder-only LLM policy $\pi_\theta$ autoregressively generates an output sequence $\mathbf{o}$, with $\mathbf{o}_t$ as the $t$-th token in $\mathbf{o}$:

$$p_\theta(\mathbf{o} \mid \mathbf{q}) = \prod_{t=1}^{T} p_\theta(o_t \mid \mathbf{q}, \mathbf{o}_{<t}). \qquad (13)$$

Within each layer $l$ and head $h \in \{1, \ldots, H\}$, multi-head self-attention computes head-specific attention maps:

$$\begin{aligned} \mathbf{S}^{(l,h)} &= \frac{\mathbf{Q}^{(l,h)}\mathbf{K}^{(l,h)\top}}{\sqrt{d_k}} + \mathbf{M}, \\ \mathbf{A}^{(l,h)} &= \mathrm{Softmax}\left(\mathbf{S}^{(l,h)}\right), \ \mathbf{O}^{(l,h)} = \mathbf{A}^{(l,h)}\mathbf{V}^{(l,h)}, \end{aligned} \qquad (14)$$

and aggregates heads via $\left[\mathrm{Concat}_{h=1}^{H} \mathbf{O}^{(l,h)}\right] \mathbf{W}_{\mathbf{O}}^{(l)}$. Causal masking enforces autoregressive dependence with $\mathbf{M}_{t,s} = \begin{cases} 0, & s \le t, \\ -\infty, & s > t, \end{cases}$. Thus,

$$\mathbf{A}_{t,s}^{(l,h)} = 0 \text{ for } s > t, \ \sum_{s \le t} \mathbf{A}_{t,s}^{(l,h)} = 1. \qquad (15)$$

Each attention map $\mathbf{A}^{(l,h)} \in \mathbb{R}^{T \times T}$ is lower-triangular, providing head-specific weights over past positions that we analyze.

## B.2  RL with Verifiable Rewards

RL optimized the LLM policy to maximize the cumulative rewards $r$ received from the verifier:

$$\max_\theta \mathcal{J}(\theta) := \mathbb{E}_{\mathbf{q} \sim Q, \mathbf{o} \sim \pi_\theta(\cdot|\mathbf{q})}[r(\mathbf{o})]. \qquad (16)$$

Policy gradient methods (Williams, 1992) are a standard approach for solving this optimization problem, with Proximal Policy Optimization (PPO) (Schulman et al., 2017) as the standard for stable, sample-efficient fine-tuning. PPO updates on data from a frozen old policy $\pi_{old}$, applies importance sampling to correct distribution shift, and optimizes a

clipped surrogate objective to constrain divergence between the new and old policies:

$$\mathcal{J}(\theta) = \mathbb{E}_{\mathbf{q} \sim Q, \mathbf{o} \sim \pi_\theta(\cdot|\mathbf{q})} \frac{1}{|\mathbf{o}|} \sum_{t=1}^{|\mathbf{o}|} \min \left( \frac{\pi_\theta(o_t|\mathbf{q}, \mathbf{o}_{<t})}{\pi_{\theta_{\text{old}}}(o_t|\mathbf{q}, \mathbf{o}_{<t})} A_t, \right.$$
$$\left. \text{clip} \left( \frac{\pi_\theta(o_t|\mathbf{q}, \mathbf{o}_{<t})}{\pi_{\theta_{\text{old}}}(o_t|\mathbf{q}, \mathbf{o}_{<t})}, 1-\epsilon, 1+\epsilon \right) A_t \right)$$
(17)

where $A_t$ is the advantage at step $t$, typically estimated via Generalized Advantage Estimation (GAE) (Schulman et al., 2015), and $\epsilon$ is a clipping hyperparameter.

Group Relative Policy Optimization (GRPO) (Shao et al., 2024) eliminates the value function (critic) and instead estimates the advantage by normalizing rewards within a group of sampled responses for the same prompt. Specifically, for a prompt $\mathbf{q}$ with $G$ responses and associated rewards $\{r_i\}_{i=1}^G$, the group-normalized advantage is given by:

$$\hat{A}_{i,t} = \frac{r_i - \text{mean}(\{r_i\}_{i=1}^G)}{\text{std}(\{r_i\}_{i=1}^G)}.$$
(18)

By emphasizing the differences among candidate outputs for the same prompt, it effectively preserves the reliability of the gradient signal, even in sparse reward settings. In addition to this modified advantage estimation, GRPO adds a KL penalty term to the clipped objective in Eq. 17.

## C  Experimental Details

### C.1  Analysis in Sec. 4

For experimental validation, we analyze attention dynamics using **Qwen3-4B-Base** (Yang et al., 2025a) on prompts from the **GSM8K** dataset (Cobbe et al., 2021a), with sampling temperature $T = 0.7$. Each prompt is constructed by first providing the system instruction: `System:  Please reason step by step, and put your final answer within \boxed{}`. We then decompose the problem into a `Context` and a `Question` field, as illustrated in Prompt C.1.

We first generate a response using standard autoregressive decoding. Once the full response is obtained, we concatenate it with the original prompt and perform a *single additional forward pass* through the model, using an eager attention implementation to extract the full attention maps via the Hugging Face `transformers` library. From this, we isolate the attention patterns corresponding to the response tokens for analysis.

In computing the Windowed Average Attention Distance (WAAD), we set the temporal window size to $W = 10$, as we are primarily interested in local attention dependencies. For the Future Attention Influence (FAI) metric, we define the influence scope to span from each token position to the end of the response.

Our analysis comprises two components: (i) For quantification analysis (e.g., Tables 1 and 2), we randomly sample 70 GSM8K problems and examine the relationship between attention-based metrics across all generated tokens; (ii) For qualitative analysis (e.g., Fig. 2), we select the problem with the shortest correct response to facilitate clear and interpretable plots. The full prompt and model response for this example are provided below:

> **Input Prompt**
>
> System: Please reason step by step, and put your final answer within \boxed{}.
> Context: Siobhan has 2 fewer jewels than Aaron. Aaron has 5 more jewels than half of Raymond's jewels.
> Question: If Raymond has 40 jewels, how many jewels does Siobhan have?

The response for analysis is:

> **Response**
>
> Step 1: Determine how many jewels Aaron has. Aaron has 5 more jewels than half of Raymond's jewels. Raymond has 40 jewels. Half of Raymond's jewels = 40 / 2 = 20. So, Aaron has 20 + 5 = 25 jewels.
> Step 2: Determine how many jewels Siobhan has. Siobhan has 2 fewer jewels than Aaron. So, Siobhan has 25 - 2 = 23 jewels.
> Final answer: Siobhan has 23 jewels.

**Experimental Details of Table 2.** Table 2 reports correlation analyses between the attention-based metrics visualized in Fig. 2. The specific protocols for computing these correlations are as follows:

- **Average Entropy of the WAAD Peaks:** We compute the average entropy at token positions identified as WAAD peaks (reported as "observed") and compare it against the average entropy over all response tokens (reported as "random"). A higher observed value indicates that WAAD peaks correspond to positions of significantly greater uncertainty.

- **Receiver & Global FAI Peak Co-occurrence:** We measure the alignment between peaks identified by receiver-head FAI (Bogdan et al., 2025) and those identified by global-head FAI. Specifically, we report the fraction of receiver FAI peaks that overlap with global FAI peaks (reported as "observed"). The "random" baseline denotes the expected overlap if receiver FAI peaks were uniformly distributed across all response positions, i.e., the proportion of Global FAI

peaks among all tokens. An increase over this baseline signifies strong structural correspondence between the two FAI variants.

- **FAI Follows/Coincides with WAAD Peak:** We evaluate the extent to which FAI peaks coincide with or immediately follow WAAD peaks. The "observed" value is the proportion of FAI peaks satisfying this temporal condition relative to all FAI peaks. The "random" baseline is the expectation by randomly shuffling FAI peak positions (preserving their count) and recomputing the overlap. A higher observed value demonstrates non-random alignment between attention activation density and forward influence.

## C.2 RL Experiments in Sec. 5

**Dataset.** For *Countdown*, we follow the data construction protocol of Pan et al. (2025) and use a test set of 512 instances (each containing four input numbers), along with a training set of 20,000 samples. The *CrossThink-QA* dataset is sourced from Akter et al. (2025). For mathematical reasoning tasks, models are trained on DAPO-Math-17K (Yu et al., 2025). The models are evaluated on five established datasets spanning a wide range of difficulty and problem types, including AIME24 and AIME25, AMC (Li et al., 2024a), MATH500 (Hendrycks et al., 2021), and Olympiad-Bench (He et al., 2024). The benchmark details are presented as follows.

**Benchmark Details.** We evaluate our method across a spectrum of reasoning tasks, ranging from relatively simple puzzles and general question-answering (QA) to challenging mathematical reasoning benchmarks. Specifically, our evaluation suite includes:

- *Countdown* (Pan et al., 2025): A symbolic reasoning puzzle in which the model is given four integers and must construct a valid arithmetic expression using basic operations $(+, -, \times, \div)$ to reach a specified target number. This task tests compositional planning and precise execution under constraints.
- *CrossThink-QA* (Akter et al., 2025): A diverse, multi-domain QA dataset that aggregates questions from heterogeneous sources (e.g., science, commonsense, and trivia), designed to assess general-purpose reasoning and knowledge integration across domains.
- *Mathematical Reasoning Benchmarks*: Five established datasets spanning a wide range of difficulty and problem types:
  - **AIME24** and **AIME25**: Collections of problems from the American Invitational Mathematics Examination (2024 and 2025) [1], featuring non-routine prob-

    lems requiring multi-step insight.
  - **AMC** (Li et al., 2024a): Problems from the American Mathematics Competitions, emphasizing logical deduction and problem-solving under time constraints.
  - **MATH500** (Hendrycks et al., 2021): A set of 500 challenging problems from the MATH dataset, covering algebra, geometry, number theory, and combinatorics.
  - **OlympiadBench** (He et al., 2024): A recent collection of international mathematical olympiad-level problems, representing the frontier of machine reasoning in mathematics.

This diverse set of tasks enables us to assess both the generalizability and the depth of our method across reasoning modalities and complexity levels.

**Plot Setup.** The performance curves are smoothed via an Exponential Moving Average (EMA): $\text{EMA}_t = \alpha \cdot \max(x_t, \text{EMA}_{t-1}) + (1 - \alpha) \cdot \text{EMA}_{t-1}$, which reflects a running average of peak performance. When the best performance is continually updated, the curve keeps rising; if performance stalls, the curve stays flat. The curve's endpoint therefore represents a relatively stable peak performance.

# D Supplementary Results

## D.1 Computational Overhead of Attention Calculation in RL

A key practical consideration for any attention-based credit assignment method in reinforcement learning (RL) is its computational overhead. While our approach requires a single additional forward pass through the frozen attention layers to compute full-sequence attention maps after autoregressive generation is complete, we emphasize that this step is fully parallelizable and incurs minimal runtime cost relative to the generation phase itself.

In standard LLM-based RL pipelines, response generation via autoregressive decoding involves hundreds to thousands of sequential forward passes (one per token), which dominates the wall-clock time of each training step. In contrast, our credit assignment operates on the already-generated sequence and performs only one batched forward pass over the concatenated prompt and response. Crucially, this pass does not interfere with the optimized inference loop used during generation; it is executed post-hoc and can be decoupled from the main actor policy.

We quantify the computational cost of our attention computation in the RL training process. After autoregressive generation, a single parallel forward pass through the frozen attention layers is performed. As shown in Table 8, this adds only 2.0-3.3% wall-clock time relative to total generation

---

[1] https://maa.org/

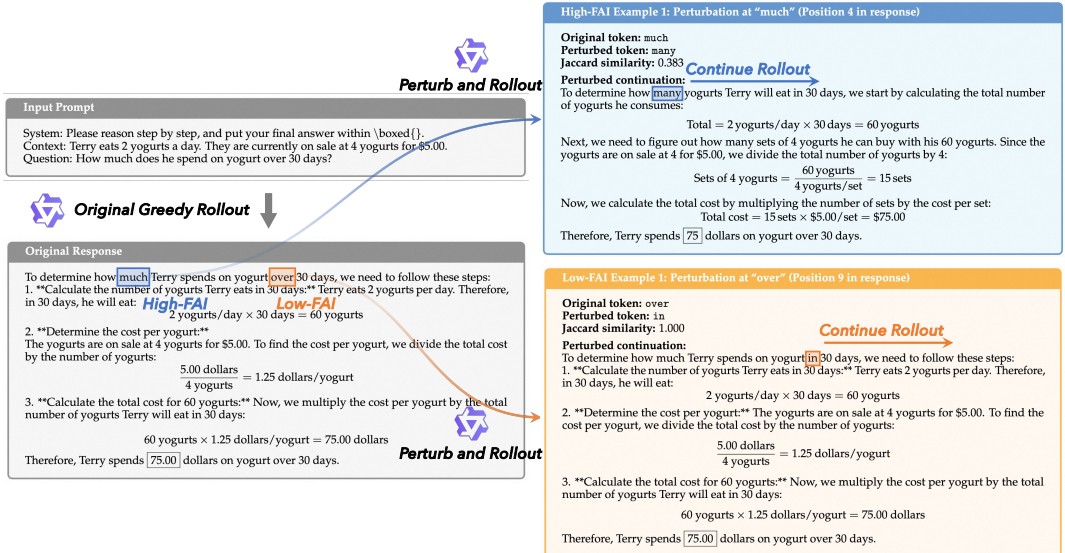

*Figure 7.* An example comparing the effects of perturbing high-FAI versus low-FAI positions and continuing the rollout of the subsequent reasoning trajectory. Perturbing a high-FAI position typically leads to a clear shift in the overall reasoning logic, whereas replacing a low-FAI token more frequently just alters local phrasing, leaving the downstream reasoning trajectory essentially unchanged.

*Table 8.* Computational overhead of credit assignment.

| Model | Seq. Length | Avg. Step Time | Attn. Credit | Relative |
|---|---|---|---|---|
| Qwen3-4B | 1,024 | 72 s (1.2 min) | 1.6 s | 2.2% |
| Qwen3-4B | 8,192 | 300 s (5 min) | 10.0 s | 3.3% |
| Qwen3-8B | 1,024 | 87 s (1.45 min) | 1.7 s | 2.0% |

time, confirming minimal overhead.

### D.2 Ablation on Layer Range Selection for Attention Signal Sources in RL

To effectively leverage internal attention patterns for credit assignment, it is crucial to identify which layers of the language model provide the most informative signals about the reasoning process. Early layers primarily encode surface-level lexical and syntactic information, while later layers capture high-level semantic and task-specific representations. We hypothesize that the *middle layers*, where abstract problem structure begins to emerge but before task-specific decoding dominates, offer the richest traces of the model's deliberative reasoning rhythm, such as bottleneck decisions or branching points between success and failure.

To test this hypothesis, we conduct an ablation study using the Coupled Rhythm Credit method with `Qwen3-4B-Base` (1K context length), comparing four variants that extract attention maps from different layer ranges: (1) **All Layers**: uniformly sampled across the entire network; (2) **Middle Layers**: five evenly spaced layers within the middle third ($\lfloor L/3 \rfloor$ to $\lfloor 2L/3 \rfloor$, where $L$ is the total number of layers); (3) **Top Layers**: the upper third of the network; and (4) **Bottom Layers**: the lower third.

As shown in Fig. 8 and Table 9, the **Middle Layers** vari-

ant consistently outperforms all others across all benchmarks, achieving the highest average score (39.2) and leading in every individual category. In contrast, using all layers dilutes the signal with less relevant early or late representations, while top and bottom layers alone miss the critical transition zone where structural reasoning crystallizes. Importantly, this layer choice should be understood as a coarse, architecture-scaled heuristic rather than a finely tuned model-specific configuration.

### D.3 Layer-Wise Analysis of Attention Patterns

To better understand how the preplan-and-anchor rhythm is distributed across the transformer stack, we conduct a layer-wise analysis on Qwen3-4B by partitioning its layers into three segments: Bottom Layers (0 to $\lfloor L/3 \rfloor$), Middle Layers ($\lfloor L/3 \rfloor$ to $\lfloor 2L/3 \rfloor$), and Top Layers ($\lfloor 2L/3 \rfloor$ to $L$). We evaluate the three core rhythmic metrics, i.e., FAI-receiver co-occurrence, WAAD peak entropy, and FAI-WAAD temporal coupling, within each segment. The results, summarized in Table 10, reveal a structured progression of attention dynamics through the network depth.

First, global influence signals emerge predominantly in the deeper layers. FAI peaks co-occur with receiver positions at 66.52% in top layers (a +296.4% lift over random), suggesting that long-range semantic anchors are established soon after input encoding and subsequently refined rather than newly created in deeper layers. The gradual decline in later layers aligns with the model's shift toward output-specific computation, where local token prediction dominates over global coherence maintenance.

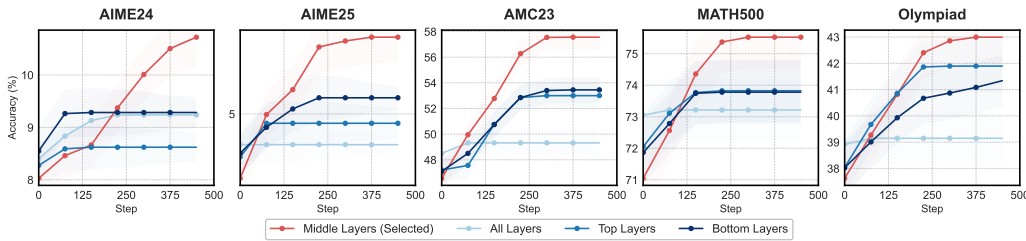

*Figure 8.* Performance curves for the Coupled Rhythm Credit method using attention signals from different layer ranges.

*Table 9.* Performance comparison the Coupled Rhythm Credit method using attention signals from different layer ranges. The final chosen setting (middle layers) is highlighted in blue. Best performance in each column is in bold.

| Method | AIME24 | AIME25 | AMC23 | MATH500 | Olympiad | Avg |
|---|---|---|---|---|---|---|
| All Layers | 9.2 | 4.7 | 49.3 | 73.2 | 39.1 | 35.1 |
| Middle Layers (Selected) | **10.7** | **5.8** | **57.4** | **75.8** | **44.1** | **39.2** |
| Top Layers | 8.6 | 4.9 | 53.0 | 73.8 | 41.9 | 36.4 |
| Bottom Layers | 9.3 | 5.2 | 53.4 | 73.8 | 41.4 | 36.6 |

*Table 10.* Attention metric linkages across different layer ranges.

| Metric | Top Layers | Middle Layers | Bottom Layers |
|---|---|---|---|
| *Receiver & Global FAI Peak Co-occurrence* | | | |
| Random | 16.78% | 22.69% | 22.58% |
| Observed | 66.52% | 51.52% | 46.33% |
| Lift | +296.42% | +127.06% | +105.18% |
| *Average Entropy of WAAD Peaks* | | | |
| Random | 0.2386 | 0.2386 | 0.2386 |
| Observed | 0.3300 | 0.2996 | 0.3634 |
| Lift | +38.31% | +25.57% | +52.31% |
| *FAI Follows/Coincides with WAAD Peak* | | | |
| Random | 35.56% | 41.08% | 40.25% |
| Observed | 56.88% | 51.69% | 54.95% |
| Lift | +59.96% | +25.83% | +36.52% |

Second, local attention structure, as measured by WAAD peak entropy, exhibits a U-shaped profile across depth. Entropy is highest in the bottom layers (+52.3% lift), indicating that local attention becomes more diverse and task-sensitive near the output, likely reflecting the generation of conclusions, numerical answers, or domain-specific phrasing. In contrast, middle layers show the lowest entropy (+25.6% lift), potentially serving as a compression stage where local dependencies are stabilized before final output formation.

Third, the functional coupling between local planning and global anchoring is strongest in top layers: 56.88% of FAI events follow or coincide with WAAD peaks in top layers (+60.0% lift). This supports our hypothesis that salient local contexts (detected via WAAD) seed subsequent global influence propagation (captured by FAI). Notably, this coupling remains above baseline even in bottom layers (+36.5% lift), suggesting that local attention continues to inform influence estimation throughout the reasoning process.

Together, these findings validate our architectural design: FAI is most informative in top-to-mid layers for identifying globally influential tokens, while WAAD gains discriminative power in later layers for fine-grained local refinement.

## D.4 Attention Pattern Generalizability to Knowledge Retrieval Tasks

Our core analysis of the preplan-and-anchor rhythm was initially developed using structured mathematical reasoning tasks (e.g., GSM8K), where reasoning proceeds in a highly procedural, step-by-step manner. A natural question is whether this rhythmic structure persists in more open-ended, knowledge-intensive settings that rely on associative retrieval rather than deductive inference. Indeed, our initial results show that performance gains on the CrossThink-QA benchmark are smaller than those on structured tasks like Countdown or math benchmarks, suggesting that the strength or manifestation of the rhythm may vary across reasoning paradigms.

To directly address this, we conducted a comparative attention analysis on *CrossThink-QA* (Akter et al., 2025), a benchmark that requires models to synthesize answers from scattered factual evidence–a setting dominated by associative rather than sequential reasoning. As shown in Table 11, the key rhythmic signatures, i.e., FAI-receiver co-occurrence, elevated WAAD peak entropy, and temporal coupling between FAI and WAAD, remain statistically significant and well above random baselines in QA. However, the lifts on WAAD peak entropy and temporal coupling between FAI and WAAD over random can be modestly reduced compared to mathematical reasoning. This aligns with the intuition that knowledge retrieval often involves multiple disjoint evidence fragments, leading to less temporally synchronized local-global dynamics.

These findings indicate that the discovered rhythm is not exclusive to symbolic reasoning but adapts to task structure: it remains detectable and functionally relevant even in associative settings, albeit with attenuated coordination. This nuanced generalizability strengthens the claim that rhythmic attention reflects a fundamental aspect of how models

organize internal computation during complex reasoning.

*Table 11.* Attention metric linkages across reasoning scenarios.

| Metric | Math | QA |
|---|---|---|
| *Receiver & Global FAI Peak Co-occurrence* | | |
| Random | 22.41% | 20.27% |
| Observed | 60.84% | 64.78% |
| Lift | +171.49% | +219.59% |
| *Average Entropy of WAAD Peaks* | | |
| Random | 0.2386 | 0.5250 |
| Observed | 0.3608 | 0.7075 |
| Lift | +51.60% | +34.76% |
| *FAI Follows/Coincides with WAAD Peak* | | |
| Random | 36.87% | 36.87% |
| Observed | 52.53% | 48.99% |
| Lift | +42.47% | +32.87% |

### D.5 Attention Pattern Generalizability to the Llama Model Architecture

A critical question is whether the attention pattern reflects a universal characteristic of transformer-based reasoning or is an artifact specific to the Qwen architecture. To address this, we replicate our attention analysis on *Llama3.1-8B* using identical hyperparameters, sequence processing protocols, and evaluation metrics.

As shown in Table 12, the core rhythmic signatures remain robust and statistically significant on Llama3.1-8B. Specifically, global influence tokens (identified via FAI) co-occur with receiver positions 72.25% of the time, nearly triple the random baseline (24.19%), yielding a +198.7% lift. Similarly, WAAD peaks exhibit measurably higher entropy (0.3082 vs. 0.2404), indicating structured local attention dynamics (+28.2% lift). Most importantly, the temporal coupling between local salience (WAAD peaks) and subsequent global influence (FAI) remains above chance, with a +34.1% lift over random alignment.

These results demonstrate that the interplay between localized planning and global anchoring is not unique to Qwen but manifests consistently across distinct transformer architectures trained on different data and with varying design choices. This strongly suggests that the preplan-and-anchor rhythm captures a regularity in how LLMs organize internal computation during complex reasoning, i.e., the alternation between context-sensitive local processing and long-range semantic propagation.

*Table 12.* Quantitative linkages between metrics on Llama3.1-8B.

| Metric Correlation | Random | Observed | Lift |
|---|---|---|---|
| Receiver & FAI Peak Co-occurrence | 24.19% | 72.25% | +198.7% |
| WAAD Peak Entropy | 0.2404 | 0.3082 | +28.2% |
| FAI Follows/Coincides with WAAD | 30.00% | 40.22% | +34.1% |

### D.6 Ablation on Attention Head Selection for Credit Assignment

The analysis in Section 4.1 reveals that attention heads exhibit distinct functional specializations: some concentrate on recent context (local-focused), while others integrate information across distant tokens (global-focused). We conduct an ablation to evaluate whether restricting credit assignment to functionally coherent head groups (identified via their average backward attention distance $d^{(l,h)}$) improves performance. Specifically, for each credit assignment method (Global-anchor, Local-chunk, and Coupled Rhythm), we compare two variants: (1) using attention maps averaged over *all heads*, and (2) using maps aggregated only over the corresponding specialized head group–i.e., $\mathcal{H}_{\text{glob}}$ for global methods and $\mathcal{H}_{\text{loc}}$ for local methods, with Coupled Rhythm leveraging a union of both.

As shown in Table 14, using targeted head groups consistently outperforms the all-heads baseline across all methods and datasets. The gains are most pronounced for Coupled Rhythm Credit, which achieves an average score of 39.2, 2.6 points higher than its all-heads counterpart. These results confirm that aligning credit assignment with the intrinsic functional organization of attention heads yields cleaner, more effective learning signals. Critically, this demonstrates that structure-aware RL benefits not only from *where* in the network to look (layer selection) but also from *which computational pathways* to prioritize (head grouping), reinforcing the value of mechanistic interpretability in guiding reinforcement learning design.

### D.7 Evaluation on Long-Horizon Planning Task

To assess the applicability of our method beyond mathematical and question-answering tasks, we evaluate on *ZebraLogic* (Lin et al., 2025), a benchmark that probes long-horizon planning through complex Constraint Satisfaction Problems (CSPs). Each instance requires the model to deduce a unique global arrangement, such as the ordering of twenty houses, from a set of interdependent, scrambled logical clues (e.g., "The Green house is immediately to the left of the White house"). Successfully solving these problems demands sustained sequential reasoning, maintenance of global consistency, and iterative refinement over extended inference trajectories.

Following the settings in (Lin et al., 2025), we train Qwen3-4B for 2 epochs with a sequence length of 1,024. As shown in Table 15, our structure-aware credit assignment methods yield substantial improvements over standard GRPO and simple baselines. Notably, the full *coupled rhythm* variant achieves 93.5% accuracy (a +9.6% absolute gain over GRPO), demonstrating that aligning reward credit with the preplan-and-anchor rhythm significantly enhances performance even in highly structured, multi-step planning scenar-

*Table 13.* Performance comparison with additional token-level credit assignment baselines on Qwen3-8B-Base with 1K response length. Bold denotes the best result.

| Method | AIME24 | AIME25 | AMC23 | MATH | Olympiad | Avg |
|---|---|---|---|---|---|---|
| GRPO | 9.3 | 7.3 | 59.1 | 77.1 | 44.2 | 39.4 |
| + random credit | 8.9 | 8.1 | 60.1 | 77.4 | 43.3 | 39.6 |
| + high-entropy credit | 8.7 | 7.7 | 60.4 | 78.0 | 45.6 | 40.1 |
| + **high-entropy selection** (Wang et al., 2025b) | 10.3 | 10.5 | 60.5 | 77.6 | 45.6 | 40.9 |
| + **CAPO** (Xie et al., 2025) | 10.5 | 10.9 | 61.1 | 78.4 | 45.9 | 41.4 |
| + **ThinkPRM-1.5B** (Khalifa et al., 2025) | 10.2 | 10.6 | 61.0 | 77.6 | 45.4 | 41.0 |
| + **AsyPPO** (Liu et al., 2025) | 9.6 | 8.2 | 59.5 | 77.8 | 44.7 | 40.0 |
| + **Reweight+Lopti** (Yang et al., 2025b) | 8.9 | 8.8 | 62.5 | 78.6 | 46.5 | 41.1 |
| + **gradient-based credit** (Green et al., 2025) | 8.7 | 9.2 | 58.4 | 78.0 | 45.4 | 40.0 |
| + **path-aggregation credit** (Minegishi et al., 2025) | 8.5 | 9.1 | 59.5 | 77.3 | 45.1 | 40.0 |
| + **token-correlation credit** (Nie et al., 2025) | 9.4 | 6.4 | 59.0 | 76.4 | 45.3 | 37.4 |
| + global-anchor credit | 10.8 (+1.4) | 11.5 (+4.2) | 63.1 (+4.0) | 78.9 (+1.8) | 46.1 (+1.9) | 42.1 (+2.7) |
| + local-chunk credit | 10.1 (+0.8) | 9.3 (+1.9) | 62.9 (+3.8) | 78.4 (+1.3) | 45.9 (+1.7) | 41.3 (+1.9) |
| + coupled rhythm credit | **11.4 (+2.1)** | **12.3 (+5.0)** | **65.4 (+6.3)** | **79.7 (+2.6)** | **47.0 (+2.8)** | **43.2 (+3.8)** |

*Table 14.* Quantitative Results of the attention-based credit assignment methods using attention maps aggregated from targeted attention heads and just by averaging all heads based on Qwen3-4B-Base.

| Method | AIME24 | AIME25 | AMC23 | MATH500 | Olympiad | Avg |
|---|---|---|---|---|---|---|
| Global-anchor Credit (All Heads) | 8.7 | 5.6 | 52.8 | 74.4 | 42.1 | 36.7 |
| Global-anchor Credit (Global-Focused Heads) | **9.3** | **5.8** | **57.6** | **75.1** | **43.0** | **38.2** |
| Local-chunk Credit (All Heads) | 8.9 | 5.2 | 52.4 | 74.1 | 42.5 | 36.6 |
| Local-chunk Credit (Local-Focused Heads) | **10.5** | **5.9** | **58.4** | **74.9** | **43.1** | **38.6** |
| Coupled Rhythm Credit (All Heads) | 8.2 | 5.8 | 53.5 | 73.4 | 41.3 | 36.5 |
| Coupled Rhythm Credit (Targeted Heads) | **10.7** | **7.8** | **57.4** | **75.8** | **44.1** | **39.2** |

ios. This result validates that the rhythmic attention pattern and credit assignment methods support coherent reasoning over extended action sequences.

*Table 15.* Accuracy (%) on the ZebraLogic long-horizon planning benchmark. Improvements over GRPO are shown in parentheses.

| Method | Accuracy (%) | Improvement |
|---|---|---|
| GRPO | 83.9 | – |
| + random credit | 89.3 | +5.4 |
| + high-entropy credit | 90.3 | +6.4 |
| + global-anchor credit | 93.2 | +9.3 |
| + local-chunk credit | 91.4 | +7.5 |
| + coupled rhythm credit | **93.5** | **+9.6** |

## D.8   Extended Baseline Comparisons on Mathematical Reasoning Benchmarks

We further compare with stronger credit-assignment baselines, including generative-PRM methods (Xie et al., 2025; Khalifa et al., 2025), value-uncertainty optimization (Liu et al., 2025), confidence-oriented reweighting (Yang et al., 2025b), entropy-based token selection (Wang et al., 2025b), and token-importance signals based on gradients, reasoning paths, or token correlations (Green et al., 2025; Minegishi et al., 2025; Nie et al., 2025). For token-scoring baselines, each method assigns reward credit to the top 40% of tokens ranked by its corresponding signal, following the same protocol used for our rhythm-aware variants.

Our structure-aware credit assignment strategies consistently outperform the newly introduced baselines. Notably,

the *coupled rhythm* variant achieves the highest performance across all datasets, with an average score of 43.2. This demonstrates that explicitly modeling the temporal and functional coupling between reasoning steps leads to more effective policy learning than treating tokens in isolation.

## D.9   Quantitative Ablation of Top-$k$ Ratio

Table 16 shows the performance of our *coupled rhythm credit* mechanism across various top-$k$ ratios, evaluated on multiple mathematical reasoning benchmarks using Qwen3-8B-Base. The results confirm that allocating credit exclusively to the top 40% of tokens yields the strongest overall performance, achieving the highest scores across all datasets and a peak average of 43.2. Both smaller ($k = 0.2$) and larger ($k \geq 0.6$) ratios lead to consistent performance drops, indicating that reinforcing too few or too many tokens dilutes the signal for critical reasoning positions.

*Table 16.* Quantitative Results of different top-$k$ ratios of coupled rhythm credit based on Qwen3-8B-Base.

| k-value | AIME24 | AIME25 | AMC23 | MATH500 | Olympiad | Avg |
|---|---|---|---|---|---|---|
| 0.2 | 10.1 | 11.3 | 58.7 | 78.0 | 45.1 | 40.6 |
| 0.4 | **11.4** | **12.3** | **65.4** | **79.7** | **47.0** | **43.2** |
| 0.6 | 9.0 | 9.9 | 59.2 | 77.7 | 45.0 | 40.2 |
| 0.8 | 10.5 | 10.6 | 58.6 | 78.1 | 45.8 | 40.7 |
| 1.0 | 8.1 | 8.6 | 60.1 | 77.5 | 44.8 | 39.8 |

## D.10   Representative Perturbation Examples

This section presents illustrative cases where token perturbations at high-FAI versus low-FAI positions lead to quali-

tatively different reasoning behaviors.

For a direct illustration, Fig. 7 first shows that replacing high-FAI tokens (even with semantically similar alternatives) frequently induces substantial shifts in the model's downstream reasoning logic, such as altering the problem-solving strategy or even changing the overall correctness, as reflected by the low Jaccard similarity. In contrast, perturbations at low-FAI positions typically affect only local phrasing while preserving the global reasoning structure, even yielding near-identical continuations. The results show that FAI reflects a reasoning-anchoring capacity that causally governs the model's planning and execution rhythm, rather than merely reflecting surface-level linguistic variation.

The specific details and examples are presented in the following. First, all examples use the same input prompt:

---

**Input Prompt**

System: Please reason step by step, and put your final answer within \boxed{}.
Context: Terry eats 2 yogurts a day. They are currently on sale at 4 yogurts for $5.00.
Question: How much does he spend on yogurt over 30 days?

---

The original response by greedy decoding is:

---

**Original Response**

To determine how much Terry spends on yogurt over 30 days, we need to follow these steps:
1. **Calculate the number of yogurts Terry eats in 30 days:** Terry eats 2 yogurts per day. Therefore, in 30 days, he will eat:

$$2 \text{ yogurts/day} \times 30 \text{ days} = 60 \text{ yogurts}$$

2. **Determine the cost per yogurt:**
The yogurts are on sale at 4 yogurts for $5.00. To find the cost per yogurt, we divide the total cost by the number of yogurts:

$$\frac{5.00 \text{ dollars}}{4 \text{ yogurts}} = 1.25 \text{ dollars/yogurt}$$

3. **Calculate the total cost for 60 yogurts:** Now, we multiply the cost per yogurt by the total number of yogurts Terry will eat in 30 days:

$$60 \text{ yogurts} \times 1.25 \text{ dollars/yogurt} = 75.00 \text{ dollars}$$

Therefore, Terry spends $\boxed{75.00}$ dollars on yogurt over 30 days.

---

We now examine perturbations at a high-FAI token, specifically the word "much" in the phrase "how much does he spend", which our FAI metric identifies as a global influence anchor due to its sustained attention from future tokens throughout the reasoning process.

---

**High-FAI Example 1: Perturbation at "much" (Position 4 in response)**

**Original token:** much
**Perturbed token:** many
**Jaccard similarity:** 0.383

**Perturbed continuation:**
To determine how many yogurts Terry will eat in 30 days, we start by calculating the total number of yogurts he consumes:

$$\text{Total} = 2 \text{ yogurts/day} \times 30 \text{ days} = 60 \text{ yogurts}$$

Next, we need to figure out how many sets of 4 yogurts he can buy with his 60 yogurts. Since the yogurts are on sale at 4 for $5.00, we divide the total number of yogurts by 4:

$$\text{Sets of 4 yogurts} = \frac{60 \text{ yogurts}}{4 \text{ yogurts/set}} = 15 \text{ sets}$$

Now, we calculate the total cost by multiplying the number of sets by the cost per set:

$$\text{Total cost} = 15 \text{ sets} \times \$5.00/\text{set} = \$75.00$$

Therefore, Terry spends $\boxed{75}$ dollars on yogurt over 30 days.

---

Despite the seemingly minor lexical substitution (both grammatically valid and semantically proximate), the resulting reasoning trajectory undergoes a profound structural reorganization. The model abandons the explicit three-step planning framework and instead adopts a set-based computational strategy: it bypasses per-unit cost calculation entirely and directly computes the number of purchasable bundles. While the final answer remains correct, the internal logic diverges significantly: the problem is reframed from a unit-cost perspective to a discrete-packaging perspective.

Nevertheless, when compared to the original response, the content-word overlap at the token level is only 0.383, indicating significant divergence in the reasoning trajectory. We observe that sometimes the change to the token may appear minor, yet the model still frequently attends back to this position internally. Thus, even when the surface semantics seem alike, it can exert a disproportionately large influence on subsequent reasoning.

**High-FAI Example 2: Perturbation at "much" (Position 4 in response)**

**Original token:** much
**Perturbed token:** long
**Jaccard similarity:** 0.255

**Perturbed continuation:**
To determine how long Terry will be able to eat yogurt for $50, we need to calculate how many yogurts he can buy with that amount of money and then figure out how many days that will last.
First, let's find out how many yogurts Terry can buy with $50. Since the yogurts are on sale at 4 for $5.00, we can calculate the cost per yogurt as follows:
Cost per yogurt = $5.00 / 4 yogurts = $1.25 per yogurt
Now, let's calculate how many yogurts Terry can buy with $50:
Number of yogurts = $50 / $1.25 per yogurt = 40 yogurts
Since Terry eats 2 yogurts a day, we can now calculate how many days he can eat yogurt for:
Number of days = 40 yogurts / 2 yogurts per day = 20 days
Therefore, Terry will be able to eat yogurt for 20 days with $50.

In this example, when we replace the token at the same position with the third-highest-probability candidate, we observe that the model's reasoning completely deviates from the correct path, leading to an incorrect final answer. When compared to the original response, the content-word overlap at the token level is only 0.255, indicating significant divergence in the reasoning trajectory.

Next, we examine the effects of perturbing low-FAI tokens by comparing the resulting reasoning trajectories after substitution. This allows us to assess how changes to less influential tokens impact the model's subsequent reasoning.

**Low-FAI Example 1: Perturbation at "over" (Position 9 in response)**

**Original token:** over
**Perturbed token:** in
**Jaccard similarity:** 1.000

**Perturbed continuation:**
To determine how much Terry spends on yogurt in 30 days, we need to follow these steps:
1. **Calculate the number of yogurts Terry eats in 30 days:** Terry eats 2 yogurts per day. Therefore, in 30 days, he will eat:

$$2 \text{ yogurts/day} \times 30 \text{ days} = 60 \text{ yogurts}$$

2. **Determine the cost per yogurt:** The yogurts are on sale at 4 yogurts for $5.00. To find the cost per yogurt, we divide the total cost by the number of yogurts:

$$\frac{5.00 \text{ dollars}}{4 \text{ yogurts}} = 1.25 \text{ dollars/yogurt}$$

3. **Calculate the total cost for 60 yogurts:** Now, we multiply the cost per yogurt by the total number of yogurts Terry will eat in 30 days:

$$60 \text{ yogurts} \times 1.25 \text{ dollars/yogurt} = 75.00 \text{ dollars}$$

Therefore, Terry spends $\boxed{75.00}$ dollars on yogurt over 30 days.

In this case, we observe a particularly interesting phenomenon: when replacing the token "over" in "over 30 days" with "in", the only change occurs at that position, and the subsequent reasoning trajectory remains completely unchanged. The Jaccard similarity is 1.00, indicating identical content-word sets. This suggests that the choice of token at this position is governed purely by local syntactic or stylistic preferences and exerts no influence on the global reasoning.

**Low-FAI Example 2: Perturbation at "spends" (Position 6 in response)**

**Original token:** spends
**Perturbed token:** will
**Jaccard similarity:** 0.946

**Perturbed continuation:**
To determine how much Terry will spend on yogurt over 30 days, we need to follow these steps:
1. **Calculate the number of yogurts Terry eats in 30 days:** Terry eats 2 yogurts per day. Therefore, in 30 days, he will eat:

$$2 \text{ yogurts/day} \times 30 \text{ days} = 60 \text{ yogurts}$$

2. **Determine the cost per yogurt:** The yogurts are on sale at 4 yogurts for $5.00. To find the cost per yogurt, we divide the total cost by the number of yogurts:

$$\frac{5.00 \text{ dollars}}{4 \text{ yogurts}} = 1.25 \text{ dollars/yogurt}$$

3. **Calculate the total cost for 60 yogurts:** Now, we multiply the cost per yogurt by the total number of yogurts Terry will eat:

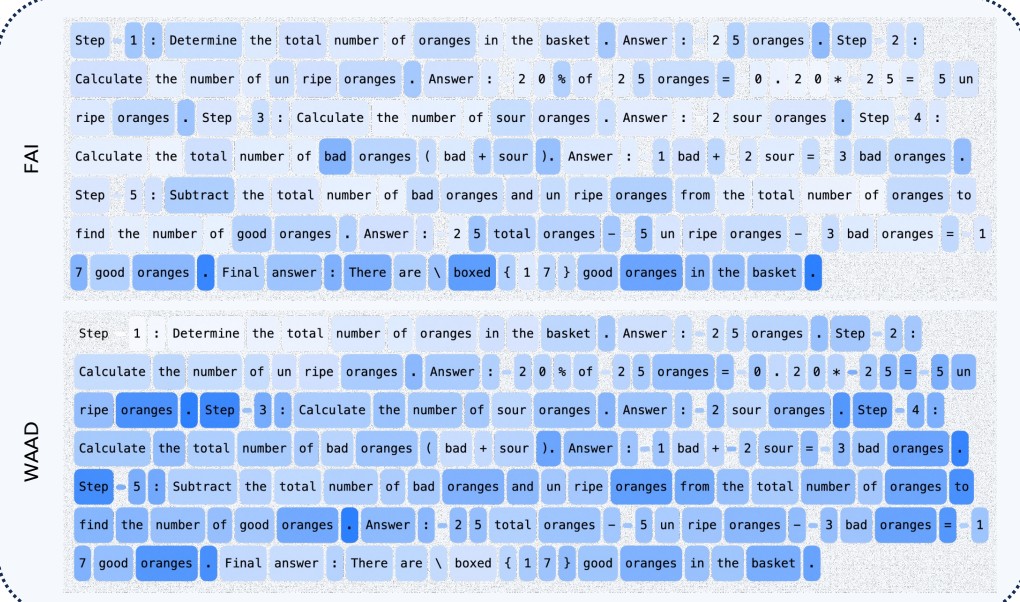

**Case 1**

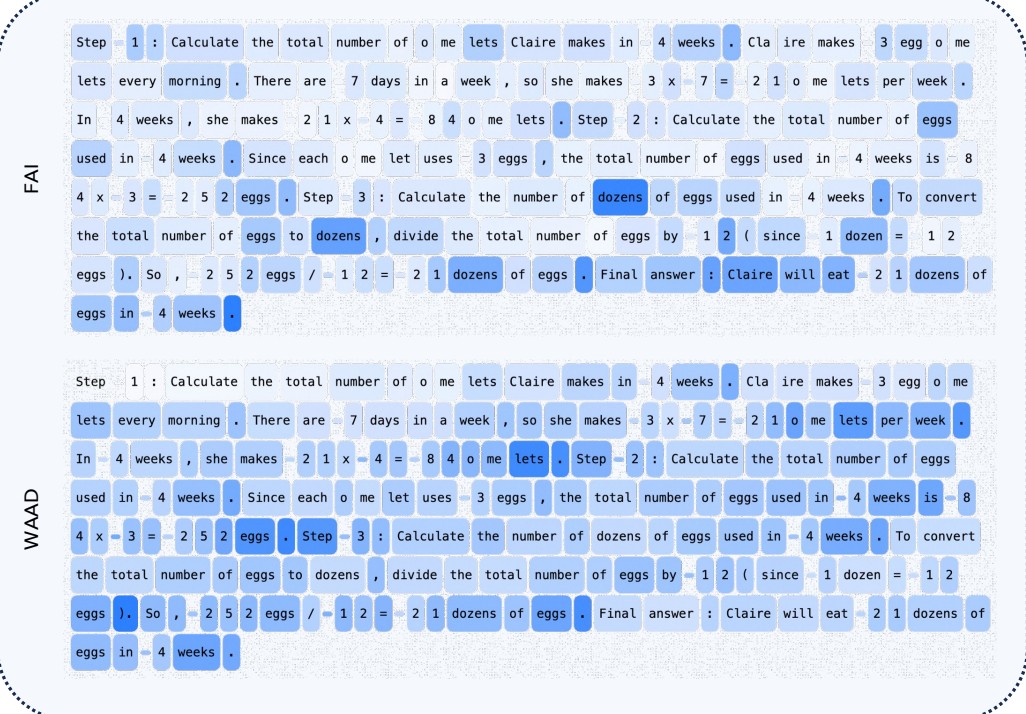

**Case 2**

*Figure 9.* Visualization cases of the token-level distributions of FAI and WAAD.

$$60 \text{ yogurts} \times 1.25 \text{ dollars/yogurt} = 75.00 \text{ dollars}$$

Therefore, Terry will spend $\boxed{75.00}$ dollars on yogurt over 30 days.

The example above exhibits the same pattern: the token replacement affects only the local phrasing, while the overall reasoning logic remains essentially unchanged. The model follows the same solution strategy, performs identical calculations, and arrives at the same final answer, confirming that such low-FAI tokens typically serve a surface-level role without altering the core inference trajectory.

*Table 17.* Full results on additional Llama-family models with 1K response length. Relative improvements are computed over GRPO within each model.

| Model | Method | AMC23 | MATH500 | Olympiad | Minerva | GSM8K | Avg. / $\Delta_{rel}$ |
|---|---|---|---|---|---|---|---|
| Llama3.2-3B | GRPO | 3.8 | 5.4 | 2.2 | 3.2 | 9.5 | 4.8 / – |
| | Random credit | 3.5 | 5.8 | 2.1 | 3.2 | 9.1 | 4.7 / −2.1% |
| | High-entropy credit | 3.8 | 5.2 | 2.3 | 3.1 | 10.0 | 4.9 / +2.1% |
| | Global-anchor credit | 4.4 | 5.9 | **3.3** | 3.2 | 10.6 | 5.5 / +14.6% |
| | Local-chunk credit | 4.0 | 5.9 | 2.4 | 2.9 | 10.4 | 5.1 / +6.3% |
| | Coupled rhythm credit | **4.8** | **6.3** | 2.9 | **3.3** | **11.0** | **5.7 / +18.8%** |
| Llama3.1-8B | GRPO | 3.2 | 8.1 | 3.2 | 5.3 | 18.7 | 7.7 / – |
| | Random credit | 4.0 | 7.8 | 3.1 | 5.2 | 17.6 | 7.5 / −2.6% |
| | High-entropy credit | 3.9 | **8.2** | 3.2 | 5.4 | 18.0 | 7.7 / +0.6% |
| | Global-anchor credit | 4.5 | 6.3 | 3.1 | 5.5 | **21.9** | 8.3 / +7.8% |
| | Local-chunk credit | 4.9 | **8.2** | 3.4 | 5.9 | 19.4 | 8.4 / +9.1% |
| | Coupled rhythm credit | **5.2** | 7.7 | **3.9** | **6.0** | 20.8 | **8.7 / +13.0%** |
| DeepSeek-R1-Distill-Llama-8B | GRPO | 42.4 | 62.8 | 28.9 | 24.3 | 78.0 | 47.28 / – |
| | Random credit | 41.6 | 62.5 | 29.4 | 24.1 | 78.2 | 47.16 / −0.25% |
| | High-entropy credit | 42.8 | 63.2 | 30.4 | 24.6 | 78.5 | 47.90 / +1.31% |
| | Coupled rhythm credit | **45.6** | **64.2** | **31.9** | **25.8** | **79.2** | **49.34 / +4.36%** |

*Table 18.* Full results on smaller Qwen3 models with 1K response length. Relative improvements are computed over GRPO within each model.

| Model | Method | AMC23 | MATH500 | Olympiad | Minerva | GSM8K | Avg. / $\Delta_{rel}$ |
|---|---|---|---|---|---|---|---|
| Qwen3-0.6B | GRPO | 8.3 | 30.5 | 7.6 | 7.3 | 49.0 | 20.54 / – |
| | Random credit | 7.9 | 31.2 | 8.0 | 6.8 | 50.1 | 20.80 / +1.27% |
| | Coupled rhythm credit | **11.2** | **37.9** | **10.1** | **12.7** | **59.7** | **26.32 / +28.14%** |
| Qwen3-1.7B | GRPO | 15.9 | 52.1 | 17.8 | 18.1 | 77.1 | 36.20 / – |
| | Random credit | 17.0 | 52.2 | 17.2 | 18.6 | 77.5 | 36.50 / +0.83% |
| | Coupled rhythm credit | **20.7** | **55.9** | **18.5** | **24.2** | **77.8** | **39.42 / +8.90%** |

## D.11 Visualization of FAI and WAAD Distributions

We visualize the token-level distributions of FAI and WAAD by directly outputting responses to randomly sampled math problems from GSM8K, computing FAI and WAAD for each token, and encoding their magnitudes via color intensity. As shown in Fig 9, tokens with high FAI typically retain intermediate reasoning results, while tokens with high WAAD often mark the beginning of a new reasoning chunk for the model. We also observe that punctuation tokens, such as commas, periods, and line breaks, consistently attract elevated attention weights across multiple layers and heads. We hypothesize that this phenomenon arises because punctuation marks appear periodically in the generated text and serve as natural syntactic and semantic boundaries. The attention mechanism, being sensitive to such regular structural cues, tends to form recurrent connections between these tokens. Thus, at these points, the model consolidates contextual evidence from preceding tokens and redistributes summarized signals to guide subsequent reasoning steps.

## D.12 Additional Results Across Model Families and Scales

To complement the compact robustness summary in Table 6, we provide the full per-benchmark results for additional model families and model scales. All experiments in this section use a 1K response length. For Llama-family base models, we report the same set of token-level credit baselines as in the main experiments. For DeepSeek-R1-Distill-Llama-8B and smaller Qwen3 models, we report the available baseline comparisons under the same evaluation protocol. Across these settings, coupled rhythm credit consistently improves over GRPO, supporting the generality of the proposed attention-derived credit signal.

## D.13 Sensitivity to WAAD/FAI Hyperparameters

We provide the full per-benchmark results for the hyperparameter sensitivity analysis summarized in Table 7(b). We vary one metric-level hyperparameter at a time while keeping the others fixed. Specifically, we ablate the WAAD window size $W$ under local-chunk credit, and the FAI horizon start $H_{lo}$ and horizon end $H_{hi}$ under global-anchor credit. The chosen values achieve the best average performance

*Table 19.* Full sensitivity results for WAAD/FAI hyperparameters on Qwen3-4B-Base with 1K response length. We vary one hyperparameter at a time under the corresponding single-signal credit strategy.

| Method | AIME24 | AIME25 | AMC23 | MATH500 | Olympiad | Avg. |
|---|---|---|---|---|---|---|
| GRPO baseline | 8.4 | 5.2 | 55.1 | 74.2 | 42.8 | 37.1 |
| *Local-chunk credit: WAAD window W* | | | | | | |
| $W = 5$ | 7.3 | 5.3 | 55.7 | 75.2 | 42.6 | 37.2 |
| $W = 8$ | 8.5 | 5.5 | 56.6 | 74.8 | 42.9 | 37.7 |
| **W = 10** (chosen) | **10.5** | **5.9** | **58.4** | 74.9 | **43.1** | **38.6** |
| $W = 12$ | 9.5 | 5.4 | 57.7 | 74.5 | 42.5 | 37.9 |
| $W = 20$ | 8.4 | 4.8 | 56.0 | 73.8 | 41.8 | 37.0 |
| *Global-anchor credit: FAI horizon start $H_{lo}$, with $H_{hi} = 100$* | | | | | | |
| $H_{lo} = 0$ | 6.0 | 4.0 | 52.1 | 71.2 | 40.7 | 34.8 |
| $H_{lo} = 5$ | 9.0 | 5.2 | 57.2 | 73.6 | 42.8 | 37.6 |
| **$H_{lo} = 10$** (chosen) | **9.3** | **5.8** | 57.6 | **75.5** | 43.0 | **38.2** |
| $H_{lo} = 15$ | 8.8 | **5.8** | 57.4 | 75.4 | 42.7 | 37.9 |
| $H_{lo} = 20$ | 7.7 | 5.4 | 56.5 | 73.8 | 42.6 | 37.2 |
| *Global-anchor credit: FAI horizon end $H_{hi}$, with $H_{lo} = 10$* | | | | | | |
| $H_{hi} = 50$ | 9.2 | 4.9 | 57.1 | 74.3 | **43.6** | 37.8 |
| **$H_{hi} = 100$** (chosen) | 9.3 | 5.8 | 57.6 | 75.5 | 43.0 | 38.2 |
| $H_{hi} = 150$ | 8.9 | 5.5 | 57.4 | 75.0 | 43.0 | 38.0 |
| $H_{hi} = 200$ | 8.9 | 5.2 | 56.7 | 74.0 | 42.9 | 37.5 |
| $H_{hi} = $ EOS | 8.3 | 5.4 | 56.9 | 74.1 | 42.5 | 37.4 |

in each sweep, while nearby alternatives generally remain above the GRPO baseline.

## D.14 Peak Alignment Patterns in Specific Examples

We analyze the positional alignment of peaks across different internal metrics, i.e., Receiver FAI, Global FAI, WAAD, and entropy, on eight representative GSM8K reasoning examples. For each pair of metrics, we compute the proportion of cases where their peaks either exactly coincide, or are offset by one position forward or backward. As shown in Table 20, several consistent patterns emerge: (i) Global FAI and WAAD peaks frequently align exactly or with WAAD leading FAI by one token, suggesting WAAD often precedes the establishment of a high-FAI anchor; (ii) Receiver FAI peaks show strong overlap with Global FAI peaks, indicating that locally attended tokens often correspond to globally salient reasoning points; and (iii) WAAD and entropy peaks also exhibit notable proximity, reflecting a potential link between contextual uncertainty and the initiation of new reasoning chunks. These fine-grained alignment trends corroborate the aggregate correlations reported in Table 20, offering token-level evidence for the coordinated dynamics among these internal signals.

*Table 20.* Comparison of peak overlaps between different metrics on randomly selected GSM8K examples.

| Comparison | Exact Match Count (Ratio) | Previous Position Count (Ratio) | Next Position Count (Ratio) |
|---|---|---|---|
| Receiver FAI Peak vs Global FAI Peak | 20 (83.3%) | 1 (4.2%) | 1 (4.2%) |
| Receiver FAI Peak vs WAAD Peak | 5 (20.8%) | 5 (20.8%) | 4 (16.7%) |
| Receiver FAI Peak vs Entropy Peak | 7 (29.2%) | 11 (45.8%) | 2 (8.3%) |
| Global FAI Peak vs WAAD Peak | 7 (25.9%) | 6 (22.2%) | 2 (7.4%) |
| Global FAI Peak vs Entropy Peak | 7 (25.9%) | 13 (48.1%) | 4 (14.8%) |
| WAAD Peak vs Entropy Peak | 12 (50.0%) | 7 (29.2%) | 2 (8.3%) |
| Receiver FAI Peak vs Global FAI Peak | 30 (56.6%) | 3 (5.7%) | 2 (3.8%) |
| Receiver FAI Peak vs WAAD Peak | 19 (35.8%) | 8 (15.1%) | 6 (11.3%) |
| Receiver FAI Peak vs Entropy Peak | 15 (28.3%) | 18 (34.0%) | 5 (9.4%) |
| Global FAI Peak vs WAAD Peak | 16 (42.1%) | 8 (21.1%) | 3 (7.9%) |
| Global FAI Peak vs Entropy Peak | 10 (26.3%) | 13 (34.2%) | 7 (18.4%) |
| WAAD Peak vs Entropy Peak | 16 (32.0%) | 10 (20.0%) | 11 (22.0%) |
| Receiver FAI Peak vs Global FAI Peak | 21 (53.8%) | 10 (25.6%) | 3 (7.7%) |
| Receiver FAI Peak vs WAAD Peak | 6 (15.4%) | 21 (53.8%) | 6 (15.4%) |
| Receiver FAI Peak vs Entropy Peak | 6 (15.4%) | 11 (28.2%) | 9 (23.1%) |
| Global FAI Peak vs WAAD Peak | 16 (38.1%) | 12 (28.6%) | 8 (19.0%) |
| Global FAI Peak vs Entropy Peak | 11 (26.2%) | 8 (19.0%) | 6 (14.3%) |
| WAAD Peak vs Entropy Peak | 12 (27.3%) | 5 (11.4%) | 3 (6.8%) |
| Receiver FAI Peak vs Global FAI Peak | 24 (58.5%) | 3 (7.3%) | 0 (0.0%) |
| Receiver FAI Peak vs WAAD Peak | 10 (24.4%) | 13 (31.7%) | 3 (7.3%) |
| Receiver FAI Peak vs Entropy Peak | 9 (22.0%) | 11 (26.8%) | 3 (7.3%) |
| Global FAI Peak vs WAAD Peak | 7 (23.3%) | 12 (40.0%) | 2 (6.7%) |
| Global FAI Peak vs Entropy Peak | 10 (33.3%) | 4 (13.3%) | 2 (6.7%) |
| WAAD Peak vs Entropy Peak | 17 (38.6%) | 5 (11.4%) | 7 (15.9%) |
| Receiver FAI Peak vs Global FAI Peak | 21 (48.8%) | 6 (14.0%) | 1 (2.3%) |
| Receiver FAI Peak vs WAAD Peak | 8 (18.6%) | 16 (37.2%) | 8 (18.6%) |
| Receiver FAI Peak vs Entropy Peak | 5 (11.6%) | 25 (58.1%) | 11 (25.6%) |
| Global FAI Peak vs WAAD Peak | 11 (34.4%) | 7 (21.9%) | 5 (15.6%) |
| Global FAI Peak vs Entropy Peak | 6 (18.8%) | 16 (50.0%) | 6 (18.8%) |
| WAAD Peak vs Entropy Peak | 14 (31.8%) | 9 (20.5%) | 4 (9.1%) |
| Receiver FAI Peak vs Global FAI Peak | 26 (56.5%) | 6 (13.0%) | 4 (8.7%) |
| Receiver FAI Peak vs WAAD Peak | 6 (13.0%) | 21 (45.7%) | 3 (6.5%) |
| Receiver FAI Peak vs Entropy Peak | 8 (17.4%) | 10 (21.7%) | 6 (13.0%) |
| Global FAI Peak vs WAAD Peak | 13 (31.7%) | 15 (36.6%) | 3 (7.3%) |
| Global FAI Peak vs Entropy Peak | 11 (26.8%) | 6 (14.6%) | 7 (17.1%) |
| WAAD Peak vs Entropy Peak | 15 (36.6%) | 4 (9.8%) | 9 (22.0%) |
| Receiver FAI Peak vs Global FAI Peak | 34 (49.3%) | 2 (2.9%) | 5 (7.2%) |
| Receiver FAI Peak vs WAAD Peak | 19 (27.5%) | 21 (30.4%) | 5 (7.2%) |
| Receiver FAI Peak vs Entropy Peak | 12 (17.4%) | 15 (21.7%) | 6 (8.7%) |
| Global FAI Peak vs WAAD Peak | 18 (40.0%) | 15 (33.3%) | 2 (4.4%) |
| Global FAI Peak vs Entropy Peak | 10 (22.2%) | 7 (15.6%) | 5 (11.1%) |
| WAAD Peak vs Entropy Peak | 20 (30.3%) | 7 (10.6%) | 8 (12.1%) |
| Receiver FAI Peak vs Global FAI Peak | 32 (71.1%) | 4 (8.9%) | 4 (8.9%) |
| Receiver FAI Peak vs WAAD Peak | 9 (20.0%) | 23 (51.1%) | 4 (8.9%) |
| Receiver FAI Peak vs Entropy Peak | 8 (17.8%) | 9 (20.0%) | 1 (2.2%) |
| Global FAI Peak vs WAAD Peak | 15 (30.6%) | 17 (34.7%) | 4 (8.2%) |
| Global FAI Peak vs Entropy Peak | 12 (24.5%) | 8 (16.3%) | 3 (6.1%) |
| WAAD Peak vs Entropy Peak | 10 (24.4%) | 9 (22.0%) | 4 (9.8%) |

