# OpenReview forum: "Attention Illuminates LLM Reasoning: The Uncovered Preplan-and-Anchor Rhythm Enables Fine-Grained Policy Optimization"
_ICML.cc/2026/Conference — ICML 2026 regular_

### Official Review · Reviewer_mN6m · 2026-03-05

**Soundness:** 3
**Presentation:** 2
**Significance:** 3
**Originality:** 2
**Overall Recommendation:** 4
**Confidence:** 3

**Summary:**

This paper addresses the challenge of sparse and uniform credit assignment in Reinforcement Learning (RL) for Large Language Models (LLMs). The authors propose using internal attention dynamics as a "mechanistic trace" to identify pivotal tokens in the reasoning process. They identify a "preplan-and-anchor rhythm" characterized by two types of attention: locally-focused heads that form phrasal chunks and globally-focused heads that identify "anchor tokens" with high downstream influence. The paper introduces two metrics, Windowed Average Attention Distance (WAAD) and Future Attention Influence (FAI), to quantify these patterns. Leveraging these, they propose three fine-grained RL strategies that reweight token-level advantages to focus learning on critical reasoning nodes. Results across math (GSM8K, AIME) and coding tasks show performance gains over standard GRPO.

**Compliance With Llm Reviewing Policy:**

Affirmed.

**Final Justification:**

This is a technically solid paper that connects mechanistic interpretability with RL by using attention-derived signals (WAAD/FAI) for fine-grained credit assignment. I find the core idea interesting and practically relevant, and the empirical results on reasoning tasks are consistently positive with low overhead. The main weaknesses are around reliance on several design choices (e.g., layer selection and metric hyperparameters) and somewhat limited evidence of broad generality, which slightly constrain the overall impact and originality. The rebuttal clarified these points and addressed my concerns, but did not substantially change my overall evaluation.

Taking everything into account (i.e., good soundness and significance, moderate originality, and improved clarity), I maintain a weak accept recommendation.

**Key Questions For Authors:**

1. Does the "middle layer" hypothesis (layers ⌊L/3⌋ to ⌊2L/3⌋) hold consistently across models with significantly different depths, such as a 70B model versus the 4B/8B models tested?
2. How were the specific window sizes for WAAD and the horizons for FAI determined? Is there a principled way to set these without manual tuning for each task?
3. If one were to use only WAAD or only FAI for credit assignment, how much of the performance gain would be preserved?

**Limitations:**

1. The "preplan-and-anchor" rhythm's dependence on the "middle third" of layers is a significant heuristic. The authors should explicitly discuss whether this pattern holds for non-transformer architectures or significantly different model scales (e.g., extremely shallow or extremely deep models).
2. The method's effectiveness is primarily demonstrated on structured reasoning tasks (math/coding). A discussion is needed on how the "rhythm" changes in more fluid, creative, or conversational contexts where logical "anchors" are less defined.
3. While the overhead is "negligible" for the models tested (2.0%–3.3%), the authors should discuss the memory and compute implications of storing and processing full attention maps for very long context sequences (e.g., 100k+ tokens), where $O(N^2)$ attention matrices become a bottleneck.

**Strengths And Weaknesses:**

**Strengths**
1. The work successfully bridges mechanistic interpretability and reinforcement learning by turning descriptive attention patterns into actionable training signals.
2. The "preplan-and-anchor" rhythm provides a clear and conceptually sound model for how LLMs may be structuring their internal reasoning into discrete semantic chunks.
3. The method is highly practical, requiring only a single additional forward pass to extract attention signals, which adds only 2.0%–3.3% computational overhead.
4. The Jaccard similarity analysis (Table 1) provides strong causal evidence that the identified "anchor" tokens are indeed more influential to the downstream reasoning trajectory than locally-dominated tokens.

**Weaknesses**
1. The effectiveness of the method is highly dependent on selecting the "middle layers" for attention signal extraction. As shown in the ablation study (Table 7), using all layers or the wrong layer range significantly dilutes the performance gains. This suggests that the method may require careful, model-specific tuning for different architectures or depths.
2. The gains are most pronounced in structured reasoning tasks like mathematics and coding. The performance on more open-ended or less "rhythmic" tasks, such as CrossThink-QA, showed smaller margins, raising questions about the generalizability of the preplan-and-anchor rhythm to general natural language generation.
3. The metrics WAAD and FAI rely on several fixed hyperparameters, such as the window size $W$ and the influence horizons $H_{lo}/H_{hi}$. The paper does not provide a comprehensive analysis of how sensitive the RL results are to these specific values across different datasets.
4. The proposed RL strategies only amplify rewards for certain tokens but do not seem to address how to handle negative advantages or provide a more holistic rethink of the reward function itself.

---

> ### Author Rebuttal · Authors · 2026-03-31
>
> ---
> > **Q1: Sensitivity to the choice of middle layers and may require model-specific tuning.**
>
> Thank you for this important question. Our use of middle layers is intended as a coarse and architecture-scaled heuristic rather than a heavily tuned, model-specific rule. Intuitionally, early layers handle local/lexical processing and late layers handle output decoding, making the middle region the best proxy for structural reasoning.
>
> We agree that this point should be framed more carefully. The paper should not imply that the "middle third" is a universal law already proved for all scales and architectures. Instead, our current evidence supports it as a practical and effective default for the tested models. We will revise the wording accordingly.
>
> We also note that the phenomenon itself is not restricted to a single architecture. We supplement the results of our method on **Llama3.1-8B and Llama3.2-3B, both with middle-layer selection**. The results show that the method and the configurations remain effective beyond Qwen, and the gains are in fact strong on Llama as well. Please refer to **Response to Reviewer 9DUd's Q2** for the results.
>
> ---
> > **Q2: Generalization beyond math/coding-style settings.**
>
> We thank the reviewer for this observation. Indeed, rhythm-aware variants consistently outperform GRPO on non-math benchmarks, including CrossThink-QA and Countdown (Table. 9, where gains are particularly strong). Furthermore, on ZebraLogic (long-horizon planning, App. D.7), coupled rhythm credit reaches 93.5% (+9.6 over GRPO).
>
> We will revise the paper to make a more nuanced claim: while the preplan-and-anchor rhythm is broadly useful for structured reasoning, the resulting RL gains naturally vary by domain. Nevertheless, the strong gains on reasoning tasks and consistent improvements on QA remain practically meaningful.
>
> ---
> > **Q3: How were hyperparameter values chosen?**
>
> Thank you for this question. Our goal was to keep the method simple and stable, so we adopt a small set of fixed defaults across all experiments rather than task-specific tuning.
>
> The choices are guided by empirical observations of attention patterns. The **WAAD window** is set to capture short-range backward consultation while avoiding absolute position biases. In practice, we observe that local attention chunks are typically within ~10 tokens (e.g., Figure 2), which motivates the window size.
>
> For **FAI**, the lower bound is aligned with the WAAD window (10), while the upper bound (50) is chosen from a range where downstream attention patterns remain relatively stable across absolute positions, allowing us to capture medium-range influence.
>
> We agree that a more exhaustive sensitivity study (e.g., for WAAD window and FAI horizon) would strengthen the paper. Please refer to **Response to Reviewer DmU9's Q2** for the results.
>
> ---
> > **Q4: Performance gains of  only using WAAD or FAI.**
>
> This is a valuable question, and the answer is already partially reflected in our reported variants. In our evaluation, **using only FAI corresponds to the global-anchor credit variant, and using only WAAD corresponds to the local-chunk credit variant** (RL results are in Table 3-4, 11, 13 and Figure 3-4, and ablation studies are in Figure 5-6 and Table 5, 12).
>
> Both single-signal variants individually improve over GRPO (e.g., on Qwen3-8B math benchmarks, lifting the 39.4 baseline to 42.1 and 41.3, respectively), but their coupled combination performs best (43.2) by capturing complementary roles.
>
> ---
> > **Q5: The method amplifies positive credit but does not address negative advantages or more global reward redesign.**
>
> Thank you for catching this. In fact, **our final implementation scales both positive and negative advantages**, ensuring both reward signals are assigned to more precise reasoning positions. The current wording reflects an earlier variant. We apologize for this inconsistency and we will correct this description in the revision.
>
> We supplement ablation confirming that while scaling either sign individually offers performance gains, **both-sign scaling achieves the best overall performance**. Please refer to **Response to Reviewer Lby9's Q4** for the results.
>
> ---
> > **Q6: Generalization to different model scales.**
>
> Thank you for this valuable question. To further examine the effect of scale, we additionally conducted experiments on **Qwen3-0.6B and Qwen3-1.7B** models. The results are as follows:
>
> |Model|Method|AMC23|MATH500|Olympiad|Minerva|GSM8K|Avg|ΔAvg|
> |-|-|-|-|-|-|-|-|-|
> |**Qwen3-0.6B**|GRPO|8.3|30.5|7.6|7.3|49.0|20.54|-|
> ||random|7.9|31.2|8.0|6.8|50.1|20.80|+1.27%|
> ||Ours|11.2|37.9|10.1|12.7|59.7|26.32|**+28.14%**|
> |**Qwen3-1.7B**|GRPO|15.9|52.1|17.8|18.1|77.1|36.20|-|
> ||random|17.0|52.2|17.2|18.6|77.5|36.50|+0.83%|
> ||Ours|20.7|55.9|18.5|24.2|77.8|39.42|**+8.90%**|
>
> These results suggest that the core pattern is not restricted to the 4B/8B regime, but can still be observed and exploited at smaller scales.

---

> > ### Author Rebuttal · Reviewer_mN6m · 2026-04-03
> >
> > Thank you for the detailed and thoughtful rebuttal. I appreciate the clarifications and additional experimental evidence you provided. The response improves clarity on several points, particularly regarding layer selection, hyperparameter choices, and additional results across models and tasks.
> >
> > Overall, I continue to find the paper technically solid and the idea interesting, and I will keep my weak accept score.

---

> > > ### Author Response · Authors · 2026-04-07
> > >
> > > Thank you for your time and for acknowledging the value of our additional analyses. Please feel free to let us know if any further questions arise, as we are always glad to help.

---

### Official Review · Reviewer_DmU9 · 2026-03-11

**Soundness:** 2
**Presentation:** 3
**Significance:** 2
**Originality:** 3
**Overall Recommendation:** 3
**Confidence:** 3

**Summary:**

This paper studies how attention patterns in LLMs can reveal how the model organizes its reasoning and how this information can improve RL training. The authors show that LLM reasoning follows a recurring "preplan-and-anchor" pattern, where the model first looks back to earlier context to plan the next step and then produces anchor tokens that guide later reasoning. Specifically, they introduce two metrics: Windowed Average Attention Distance (WAAD) and Future Attention Influence (FAI) to identify these tokens and use them to reweight per-token advantages in on-policy RL. Across multiple tasks such as puzzles, QA, and math reasoning, the method outperforms random token selection and entropy-based baselines.

**Compliance With Llm Reviewing Policy:**

Affirmed.

**Final Justification:**

The authors addressed some of my concerns. however, some issues remain and require further clarification. The paper needs a clearer explanation of the advantages and differences compared to existing methods for token importance and credit assignment. Additionally, more justification is needed for the WAAD/FAI signals and for the choice of hyperparameters, especially given that the proposed method relies on a large number of them.

**Key Questions For Authors:**

* How stable are WAAD/FAI patterns post-training? Do the attention shift as the model improves, and does that affect the reliability of token selection when doing RL?

**Limitations:**

yes

**Strengths And Weaknesses:**

## Strengths:
* The paper introduces simple and intuitive metrics that quantify local and global attention behaviors.
* The evaluation covers multiple settings and benchmarks.
* The paper is generally well-written and easy to follow.

## Weaknesses:
* The main method relies on attention weights as faithful indicators of causal influence. This questions the generality of the approach, since attention scores are not generally reliable explanations of model behavior and may not generalize across different models. The claim of a general preplan-and-anchor rhythm is only analyzed on a single base model and a limited sample.
* The method introduces several thresholds/hyperparameters with limited sensitivity analysis.
* The method requires computing full attention maps during training to obtain WAAD and FAI.
* The paper lacks comparison with more advanced token-level credit assignment [1] or token-level RM [2] methods.

[1] Xie, Guofu, et al. "CAPO: Towards Enhancing LLM Reasoning through Generative Credit Assignment." arXiv preprint arXiv:2508.02298 (2025).

[2] Khalifa, Muhammad, et al. "Process reward models that think." arXiv preprint arXiv:2504.16828 (2025).

---

> ### Author Rebuttal · Authors · 2026-03-31
>
> We thank the reviewer for the thoughtful comments and for recognizing the paper’s clarity, the intuitive nature of the proposed metrics, and the breadth of the evaluation. Below we address the main concerns in turn.
>
> ---
> > **Q1: Limited generality (single model and limited analysis)**
>
> We agree that attention should not be treated as a perfect or exhaustive explanation of model behavior. Our intended claim is narrower: attention-derived signals such as WAAD and FAI are **useful internal proxies** for identifying positions that are structurally important to the reasoning trajectory, and these proxies can improve token-level credit assignment in RL.
>
> We supplement results for the generality evaluation:
>
> **Generality across different models.** Indeed, we have replicated the same analysis on **Llama3.1-8B** using the same setup in Appendix D.5, where the core patterns remain statistically significant. These results presented in **https://anonymous.4open.science/r/Attn-6B42/tables.pdf Table 1** suggest that the core phenomenon is not specific to Qwen.
>
>
> We have also additionally evaluated our method on **Llama 3.1-8B** and **Llama 3.2-3B** under the same RL setting. The results in **https://anonymous.4open.science/r/Attn-6B42/tables.pdf Table 2** show that the proposed credit assignment strategy remains effective beyond Qwen, and the gains are in fact stronger on Llama.
>
> **Generality across samples.** Indeed, our analysis is not based on isolated examples. We sample 70 GSM8K problems and quantify metric linkages (Tables 1, 2, 10), while Figure 2 serves only as an illustrative visualization. The reported patterns are statistically validated across samples.
>
> ---
> > **Q2: Sensitivity to thresholds / hyperparameters.**
>
> The method introduces a small number of hyperparameters (WAAD window, FAI horizon, top-k ratio), which we intentionally keep **fixed across tasks** rather than heavily tuned. Among them, the most influential is the **top-k ratio**. As shown in Table 14, top 40% achieves the best performance, while both smaller and larger values degrade results.
>
> We agree that a more exhaustive sensitivity study (e.g., for WAAD window and FAI horizon) would strengthen the paper. As shown in the table in **https://anonymous.4open.science/r/Attn-6B42/tables.pdf Table 4**, we evaluate the impact of different hyperparameter choices, including **(1) WAAD window size**, **(2) FAI horizon start** and **(3) FAI horizon end**. While performance naturally peaks at our selected settings, the method maintains consistent improvements over the baseline across a reasonable range of variations.
>
> ---
> > **Q3: Overhead concerns.**
>
> Our overhead analysis is presented in Appendix D.1. Below we provide a brief summary.
>
> Our method is performed **after** the actor_infer (using FlashAttention) has completed autoregressive generation of the full response . At this stage, we concatenate the original prompt and the generated response into a single sequence and perform **one additional forward pass** through the frozen attention layers of actor_attn (raw Attention) to extract attention maps.
>
> Crucially:
>
> - Generation already requires **hundreds to thousands** of sequential forward passes.
> - Our method requires **only a single, fully parallelizable** forward pass over the complete sequence.
> - The additional overhead is around 2-3% (even without FlashAttention): **https://anonymous.4open.science/r/Attn-6B42/tables.pdf Table 3**.
>
> ---
> > **Q4: Comparison with CAPO / process reward models**
>
> We supplement comparisons to **CAPO [1]** and **ThinkPRM [2]** on Qwen3-8B in **https://anonymous.4open.science/r/Attn-6B42/tables.pdf Table 5**.  Both methods serve as strong baselines that noticeably improve upon GRPO. Nevertheless, our method still achieves further improvements over these baselines.
>
> ---
> > **Q5: Attention shifts as the model improves in RL**
>
> WAAD and FAI are computed online from the current policy, so the high-WAAD or high-FAI token subsets adapt dynamically as the model evolves. **Our empirical analysis also suggests that WAAD and FAI may serve as stable, intrinsic signals within LLMs.** Specifically, on Qwen3-4B (1K context), we collected greedy MATH-500 responses from the initial model, then trained with standard GRPO or GRPO + Coupled Rhythm Credit, saving checkpoints every 100 steps. At each checkpoint, we identified the top-40% WAAD and FAI tokens on the same input sequences and measured their overlap with Step 0. The results in **https://anonymous.4open.science/r/Attn-6B42/tables.pdf Table 6** show that WAAD and FAI patterns remain surprisingly stable across training.
>
> ---
>
> We hope this response helps address the reviewer’s concerns. We appreciate the reviewer’s comments, which helped us improve the framing of the method’s reliability, practicality, and relation to prior work.
>
> [1] CAPO: Towards Enhancing LLM Reasoning through Generative Credit Assignment. 2025.
>
> [2] Process reward models that think. 2025.

---

> > ### Author Rebuttal · Reviewer_DmU9 · 2026-04-03
> >
> > Thank the authors for their response. It has addressed some of my concerns. However, I still have some concerns regarding the method. My primary remaining issue is whether WAAD/FAI are just useful correlations or actually the right signals for credit assignment. When do these proxies fail, and can they mislead RL when attention is spurious or stylistic and not reasoning-relevant? Moreover, I think the paper still needs to better clarify what is unique about attention-based credit shaping relative to other token-importance signals. Finally, the framework relies on heuristics and hyperparameter tuning (layer selection, WAAD window size, quantile thresholds, etc.) without a principled method for choosing these settings.
> >
> > Therefore, I will maintain my score.

---

> > > ### Author Response · Authors · 2026-04-04
> > >
> > > We thank the reviewer for the thoughtful comments and the opportunity to clarify these points. We also apologize that our previous response devoted too much space to supplementary results while not sufficiently explaining the broader role and positioning of the method.
> > >
> > > ------
> > >
> > > > **Q1. Whether WAAD/FAI are just useful correlations or actually the right signals for credit assignment.**
> > >
> > > We do not view attention as a complete explanation of model behavior. Our claim is narrower: WAAD and FAI are useful internal proxies for identifying structurally important positions in the reasoning trajectory, and these signals can improve token-level credit assignment in RL.
> > >
> > > The paper has two connected contributions: attention analysis and RL design. The analysis matters precisely because raw attention is noisy and heterogeneous. The analysis component of the paper serves as a **filtering and refinement process**: rather than trusting attention directly, we extract statistically reliable patterns and filter out irrelevant activity through head selection and local/global coupling analysis.
> > >
> > > These extracted signals show statistically meaningful alignment with externally validated behavior. For example, Table 2 shows that the discovered patterns correspond to measurable external effects, while Table 1 and Figure 7 show that perturbing high-FAI positions drastically alters trajectories. The downstream RL results further support that the filtered attention patterns are useful for optimization.
> > >
> > > > **Q2. When do these proxies fail, and can they mislead RL when attention is spurious or stylistic and not reasoning-relevant?**
> > >
> > > Our results suggest the method is not limited to a single task or model family, though the magnitude of gains varies.
> > >
> > > On the **analysis side**, the core coupling patterns remain significantly above chance in QA, although weaker than in math, and we replicated the same phenomenon on Llama3.1-8B. On the **RL side**, rhythm-aware variants consistently outperform GRPO on non-math tasks such as CrossThink-QA, Countdown, and ZebraLogic, and additional experiments on Llama3.1-8B and Llama3.2-3B (in the supplementary link) show that the credit assignment strategy remains effective beyond Qwen.
> > >
> > > We believe this generality is explainable. The underlying attention behaviors (the tendency to generate in local chunks, repeatedly attend back to key positions, and their temporal coupling) remain significant across settings because they seem to reflect broad inductive habits of the model’s attention dynamics, likely shaped during pretraining and generation.
> > >
> > > ------
> > >
> > > > **Q3. Better clarify what is unique relative to other token-importance signals.**
> > >
> > > A practical difference is that many token-importance methods rely on a separately trained scorer, which introduces extra supervision, model complexity, and potential generalization issues as the policy changes during RL. In practice, accurately estimating process-level rewards is challenging and non-trivial. In contrast, our method derives signals directly from the current policy’s own attention structure, requiring no auxiliary model, no extra labels, and only one additional forward pass.
> > >
> > > More importantly, our method does not attempt to judge the local correctness of each intermediate step as PRM does. Instead, it redistributes sequence-level reward toward tokens that are most influential in routing information to the final answer (whether correct or wrong). Our role is to make sequence-level RL signals more targeted by concentrating updates on the tokens that most strongly shaped the produced outcome and fix them more quickly.
> > >
> > > Supplementary comparisons to CAPO and ThinkPRM further support that this internal attention-based signal provides complementary value (Table 5 in the supplementary link).
> > >
> > > ------
> > >
> > > > **Q4. Reliance on heuristics and hyperparameter tuning without a principled method for choosing these settings.**
> > >
> > > Our first choices are guided by empirical observations of attention patterns. The WAAD window is set to capture short-range backward consultation while avoiding absolute position biases. In practice, we observe that local attention chunks are typically within ~10 tokens (e.g., Figure 2), which motivates the window size. For FAI, the lower bound is aligned with the WAAD window (10), while the upper bound is chosen to stably capture medium-range influence.
> > >
> > > Importantly, these settings were chosen **once** in the Qwen3-4B, 1K response-length setup and were **not retuned** across model sizes, families, or response lengths. Their continued effectiveness across settings suggests that the method is reasonably robust rather than heavily dependent on per-case tuning.
> > >
> > > ------
> > >
> > > We hope this clarification helps address the reviewer’s remaining concerns and better situates the paper in its broader context. We thank the reviewer again for the thoughtful feedback, which has helped us substantially improve both the explanation and the positioning of the paper.

---

### Official Review · Reviewer_Lby9 · 2026-03-12

**Soundness:** 2
**Presentation:** 3
**Significance:** 3
**Originality:** 3
**Overall Recommendation:** 4
**Confidence:** 4

**Summary:**

This paper investigates LLM attention dynamics to derive fine-grained credit assignment signals for Reinforcement Learning (RL). The authors propose two metrics: WAAD (Windowed Average Attention Distance) to capture local look-back patterns and FAI (Future Attention Influence) to measure a token's impact on downstream generation. Based on an observed "preplan-and-anchor" rhythm—where "preplan" tokens (high WAAD) precede "anchor" tokens (high FAI)—the paper introduces three RL credit reweighting strategies (Local, Global, and Coupled). Experiments on Countdown, CrossThink-QA, and math benchmarks show consistent improvements over the GRPO baseline.

**Compliance With Llm Reviewing Policy:**

Affirmed.

**Key Questions For Authors:**

# Questions
1. The effectiveness of the "rhythm" appears strongest in procedural/mathematical reasoning. Could the authors provide experimental results demonstrating its performance on a wider range of task types?
2. Currently, reweighting only applies to non-negative advantages. Does applying these weights to negative advantages impact training stability or performance?
3. Why was Jaccard similarity of content words chosen for counterfactual analysis? Would a semantic or logical consistency metric provide a more accurate measure of trajectory deviation?

**Limitations:**

1. Presenting several specific examples, aligning the WAAD peak, FAI peak, and natural language inference steps one by one, will help readers determine whether the explanation is truly credible.
2. I suggest the author state more explicitly in the narrative: WAAD/FAI is currently a useful proxy for the reasoning process, rather than a rigorously proven decomposition of true reasoning.

**Strengths And Weaknesses:**

# Strengths
1. The paper successfully bridges mechanistic interpretability with RL optimization, using internal model signals to address the coarse-grained nature of standard RLVR/GRPO credit assignment.
2. WAAD and FAI offer clear, quantifiable proxies for "local planning" and "global anchoring," providing a structured way to identify critical reasoning steps.
3. The method shows stable performance gains over GRPO across various model scales (Qwen3-4B/8B) and tasks, supported by meaningful ablations on token-selection strategies.
# Weaknesses
1. The effectiveness of the "rhythm" appears strongest in procedural/mathematical reasoning. The authors acknowledge weaker gains in knowledge-retrieval tasks, suggesting the method's universality across diverse LLM applications remains unproven.
2. While the paper shows that high-FAI tokens influence future rollouts, the "preplan-and-anchor" rhythm is described as an "intrinsic reasoning mechanism." This causal claim is overstated; these metrics may currently be useful proxies rather than the definitive internal logic of the model.
3. Comparisons are limited to GRPO and simple random/entropy-based reweighting. The paper would be more convincing if compared against stronger baselines, such as those using value-uncertainty or intermediate verifier-derived heuristics for credit shaping.

---

> ### Author Rebuttal · Authors · 2026-03-31
>
> We thank the reviewer for the careful reading and encouraging feedback. We especially appreciate the recognition that our work helps connect mechanistic interpretability with RL optimization. Below we address the main concerns.
>
> ---
> > **Q1: Generality beyond mathematical reasoning**
>
> Thank you for this important question. Indeed, both our analysis results and our RL results suggest that the proposed mechanism is not limited to mathematics.
>
> **(1) Analysis side**: In QA settings, core couplings in Table 9 remain **significantly above chance**. Though weaker than in math, this indicates rhythm is a general internal pattern.
>
> **(2) RL side**: Rhythm-aware variants consistently outperform GRPO on non-math benchmarks, including **CrossThink-QA** and **Countdown** (where gains are particularly strong). Further, on **ZebraLogic** (long-horizon planning, App. D.7), coupled rhythm credit reaches 93.5% (+9.6 over GRPO).
>
> ---
> > **Q2: The paper may overstate the rhythm as an intrinsic reasoning mechanism, whereas WAAD/FAI may be better understood as useful proxies.**
>
> We agree and appreciate this suggestion. We do not claim WAAD/FAI provide a complete or uniquely correct decomposition of reasoning, but rather useful internal proxies. We will revise the text to explicitly reframe the rhythm as a **model-internal regularity and useful proxy for credit assignment**, rather than a definitive causal theory of reasoning.
>
> ---
> > **Q3: The baseline comparisons are limited, and the paper would be more convincing with stronger related baselines.**
>
> Thank you for this very helpful suggestion. We have expanded our comparisons accordingly, specifically including the value-uncertainty-based **AsyPPO** [1] and the verifier-derived heuristic approach, **CAPO** [2]. Indeed, we have added three additional token-weighting baselines in Table 11: **gradient-based [3], path-aggregation [4], and token-correlation [5] weighting**. These additions provide a broader comparison against representative token-level scoring strategies.
>
> |Method|AIME24|AIME25|AMC23|MATH|Olympiad|Avg|
> |-|-|-|-|-|-|-|
> |GRPO|9.3|7.3|59.1|77.1|44.2|39.4|
> |+ **value-uncertainty method (AsyPPO)** [1]|9.6|8.2|59.5|77.8|44.7|40.0|
> |+ **verifier-derived heuristics (CAPO)** [2]|10.5|10.9|61.1|78.4|45.9|41.4|
> |+ gradient-based credit [3]|8.7|9.2|58.4|78.0|45.4|40.0|
> |+ path-aggregation credit [4]|8.5|9.1|59.5|77.3|45.1|40.0|
> |+ token-correlation credit [5]|9.4|6.4|59.0|76.4|45.3|37.4|
> |+ coupled rhythm credit|**11.4**|**12.3**|**65.4**|**79.7**|**47.0**|**43.2**|
>
> ---
> > **Q4: Why apply reweighting only to non-negative advantages?**
>
> Thank you for catching this. In fact, **our final implementation scales both positive and negative advantages**, ensuring both reward signals are assigned to more precise reasoning positions. The current wording reflects an earlier variant. We apologize for this inconsistency and we will correct this description in the revision.
>
> The ablation below (Qwen3-8B, 1K context, coupled rhythm credit) confirms that while scaling either sign individually offer performance gains, **both-sign scaling provides the most comprehensive training signal and achieves the best overall performance**.
>
> |Method|AIME24|AIME25|AMC23|MATH|Olympiad|Avg|
> |-|-|-|-|-|-|-|
> |GRPO|9.3|7.3|59.1|77.1|44.2|39.4|
> |positive-only scaling|10.6|10.9|64.5|77.2|46.6|42.0|
> |negative-only scaling|9.8|8.4|62.2|77.9|45.7|40.8|
> |both-sign scaling|**11.4**|**12.3**|**65.4**|**79.7**|**47.0**|**43.2**|
>
> ---
> > **Q5: Use of Jaccard similarity**
>
> Thank you for this valuable question. For the counterfactual metric, we chose Jaccard overlap of content words as a conservative measure of structural deviation because it is more sensitive to fine-grained changes in reasoning trajectories than coarse-grained semantic metrics.
>
> However, we agree that semantic-similarity metrics are valuable complements. Using the same intervention setup, we computed the cosine similarity of the original and counterfactual continuations using a Qwen3-4B sentence embedding model. As shown below, **perturbing top-k positions still results in larger semantic deviations than perturbing bottom-k positions**.
>
> |Metric|Value|
> |-|-|
> |Mean Similarity (Top-k perturbed)|0.84|
> |Mean Similarity (Bottom-k perturbed)|0.90|
> |% Cases where Top-k causes larger deviation|82.69\%|
>
> ---
>
> We hope this response addresses the reviewer’s concerns. We are grateful for the suggestions, which helped us sharpen both the scope claims and the empirical support of the paper.
>
> [1] Asymmetric Proximal Policy Optimization: mini-critics boost LLM reasoning.
>
> [2] CAPO: Towards Enhancing LLM Reasoning through Generative Credit Assignment.
>
> [3] Contextual gradient recomposition for sequential coherence preservation in large language model token generation.
>
> [4] Topology of Reasoning: Understanding Large Reasoning Models through Reasoning Graph Properties.
>
> [5] A text is worth several tokens: Text embedding from llms secretly aligns well with the key tokens.

---

> > ### Author Rebuttal · Reviewer_Lby9 · 2026-04-03
> >
> > Thank you for the rebuttal. I find this work to be a meaningful contribution. The additional analyses provided by the authors will help future readers better understand and contextualize the paper. I will keep my score unchanged.

---

> > > ### Author Response · Authors · 2026-04-07
> > >
> > > Thank you for your constructive feedback and for recognizing our work as a meaningful contribution. We would be more than happy to address any remaining questions you might have.

---

### Official Review · Reviewer_9DUd · 2026-03-13

**Soundness:** 2
**Presentation:** 3
**Significance:** 2
**Originality:** 3
**Overall Recommendation:** 3
**Confidence:** 4

**Summary:**

This paper argues that current RL algorithms typically assign uniform credit across an entire generated sequence, which blurs the distinction between pivotal reasoning steps and routine tokens. To address this limitation, the authors leverage attention maps to identify critical tokens during generation and introduce two metrics for it. Their analysis reveals two key phenomena in the model’s reasoning dynamics: local phrasal processing and global contextual anchoring. Based on these findings, the paper proposes a method with three variants to rescale the advantage of critical tokens. Experimental results demonstrate consistent performance gains across a range of reasoning tasks.

**Compliance With Llm Reviewing Policy:**

Affirmed.

**Final Justification:**

I will maintain my score. The experiments added in the rebuttal are limited in terms of model family, considering the rebuttal period is enough for experiments on gemma-4, ministral-3, minicpm-4.1, etc. The additional experiments are not clearly setup and compared with baselines. The paper can be better improved for the next submission.

**Key Questions For Authors:**

See weakness

**Limitations:**

Yes

**Strengths And Weaknesses:**

Strengths

1. The paper is well-written and easy to understand.
2. The analysis in Section 4 is interesting. In particular, the first step of grouping attention heads by behavior and then analyzing token-level behavior is an insightful approach and may provide useful ideas for future work.
3. The proposed method is simple and appears easy to incorporate into existing RL algorithms.

Weaknesses

1. Potential bias from the model. I am concerned about using attention patterns learned by the model prior to RL as a training signal. The method may amplify existing biases in attention patterns rather than discovering meaningful reasoning structures. It might also encourage the model to optimize attention artifacts instead of improving reasoning quality. The paper does not analyze or discuss this potential risk.
2. Limited experimental scope. The experiments are conducted only on Qwen3 models. This raises some concerns because prior work [1] has reported issues with Qwen2.5. Therefore, I worry that similar issues may also appear in Qwen3, especially when combined with the potential bias mentioned in the first weakness. The authors should evaluate their method on a broader range of models.
3. The set of baselines is limited. Several prior works study critical tokens or propose techniques closely related to the motivation of this paper, but they are not experimentally or conceptually compared with the proposed method (e.g., [2] and [3]).

----
[1] Spurious Rewards: Rethinking Training Signals in RLVR

[2] Beyond the 80/20 Rule: High-Entropy Minority Tokens Drive Effective Reinforcement Learning for LLM Reasoning

[3] Do Not Let Low-Probability Tokens Over-Dominate in RL for LLMs

---

> ### Author Rebuttal · Authors · 2026-03-31
>
> We thank the reviewer for the thoughtful feedback and for recognizing the clarity of the paper, the insights in Section 4, and the practicality of the proposed RL recipe.
>
> ---
> > **Q1: Potential bias from attention patterns.**
>
> Thank you for raising this point. To clarify, our method does not optimize attention patterns, nor does it encourage the model to imitate or amplify specific attention artifacts. The optimization objective remains the same RL reward as GRPO. Attention-derived signals (WAAD and FAI) are used only to rescale token-level advantages, i.e., **as observational signals for credit assignment rather than optimization targets**. The model cannot directly optimize toward them. Instead, they guide where sequence-level reward should be propagated more strongly.
>
> Empirically, these signals are not merely superficial artifacts. Perturbing high-FAI positions drastically alters trajectories (reducing rollout overlap in 87.14% of cases; Table 1, Fig. 7), confirming their functional tie to reasoning; RL experiments (Tables 3-4, Figs. 3-4) further validate this connection.
>
> We do agree that pretrained attention patterns should not be treated as unbiased or causal ground truth for reasoning. In the revision, we will clarify that attention patterns are not assumed to be unbiased explanations of reasoning; rather, they provide useful internal signals to identify structurally important positions and improve token-level credit assignment.
>
> ---
> > **Q2: Limited experimental scope, especially only on Qwen-family models.**
>
> Thank you for this suggestion. We agree that demonstrating robustness beyond the Qwen family is important, and we provide supplementary results for both **(1) RL and (2) further analysis**.
>
> **RL on Llama families.** we evaluate our method on **Llama 3.1-8B** and **Llama 3.2-3B** under the same RL setting. The results below show that the proposed credit assignment strategy remains effective beyond Qwen, and the gains are in fact strong on Llama as well:
>
> |Model|Method|AMC23|MATH500|Olympiad|Minerva|GSM8K|Avg|ΔAvg(%)|
> |:-|:-|-|-|-|-|-|-|-|
> |**Llama3.2-3B**|GRPO|3.8|5.4|2.2|3.2|9.5|4.8|-|
> ||Random|3.5|5.8|2.1|3.2|9.1|4.7|-2.1%|
> ||High-entropy|3.8|5.2|2.3|3.1|10.0|4.9|+2.1%|
> ||Global-anchorCredit|4.4|5.9|3.3|3.2|10.6|5.5(+0.7)|**+14.6%**|
> ||Local-chunk Credit|4.0|5.9|2.4|2.9|10.4|5.1(+0.3)|**+6.3%**|
> ||Coupled Rhythm Credit|4.8|6.3|2.9|3.3|11.0|5.7(+0.9)|**+18.8%**|
> |**Llama3.1-8B**|GRPO|3.2|8.1|3.2|5.3|18.7|7.7|-|
> ||Random|4.0|7.8|3.1|5.2|17.6|7.5|-2.6%|
> ||High-entropy|3.9|8.2|3.2|5.4|18.0|7.7|+0.6%|
> ||Global-anchor Credit|4.5|6.3|3.1|5.5|21.9|8.3(+0.6)|**+7.8%**|
> ||Local-chunk Credit|4.9|8.2|3.4|5.9|19.4|8.4(+0.7)|**+9.1%**|
> ||Coupled Rhythm Credit|5.2|7.7|3.9|6.0|20.8|8.7(+1.0)|**+13.0%**|
>
> **Analysis experiments.** We replicate our analysis on Llama3.1-8B, observing consistent qualitative patterns:
>
> - **Receiver & Global FAI Peak Co-occurrence**: 72.25% (vs. 24.19% random, +198.68%)
> - **WAAD Peak Entropy**: +28.20% over random
> - **FAI alignment with WAAD peaks**: +34.07% over random
>
> These results reduce the concern that our method relies on Qwen-specific behaviors. Rhythm-aware credit assignment appears to generalize across architectures.
>
> ---
> > **Q3: Limited baseline comparisons.**
>
> Thanks for the suggestion. We supplement comparisons to these works [1, 2] on Qwen3-8B as follows. Indeed, we have also incorporated three additional token-weighting methods in Table 11, including **gradient-based [3], path-aggregation [4], and token-correlation [5] weighting**. These additions strengthen the empirical positioning of our method relative to existing strategies.
>
> |Method|AIME24|AIME25|AMC23|MATH|Olympid|Avg|
> |:-|-|-|-|-|-|-|
> |GRPO|9.3|7.3|59.1|77.1|44.2|39.4|
> |+ **high-entropy selection [1]**|10.3|10.5|60.5|77.6|45.6|40.9|
> |+ **Reweight+Lopti [2]**|8.9|8.8|62.5|78.6|46.5|41.1|
> |+ **gradient-based credit[3]**|8.7|9.2|58.4|78.0|45.4|40.0|
> |+ **path-aggregation credit[4]**|8.5|9.1|59.5|77.3|45.1|40.0|
> |+ **token-correlation credit[5]**|9.4|6.4|59.0|76.4|45.3|37.4|
> |+ global-anchor credit|10.8|11.5|63.1|78.9|46.1|42.1|
> |+ local-chunk credit|10.1|9.3|62.9|78.4|45.9|41.3|
> |+ coupled rhythm credit|11.4|12.3|65.4|79.7|47.0|**43.2**|
>
>
> ---
> We hope this response could help address your concerns. We believe that this work contributes to this community and we are more than happy to address any further concerns you may have.
>
> [1] Beyond the 80/20 Rule: High-Entropy Minority Tokens Drive Effective Reinforcement Learning for LLM Reasoning
>
> [2] Do Not Let Low-Probability Tokens Over-Dominate in RL for LLMs
>
> [3] Contextual gradient recomposition for sequential coherence preservation in large language model token generation. 2025.
>
> [4] Topology of Reasoning: Understanding Large Reasoning Models through Reasoning Graph Properties. Neurips 2025.
>
> [5] A text is worth several tokens: Text embedding from llms secretly aligns well with the key tokens. ACL 2025.

---

> > ### Author Rebuttal · Reviewer_9DUd · 2026-04-03
> >
> > Thank you for your detailed response. The results in the tables seem too low. Are there any evidences or experimental results from other papers that can support these observations?

---

> > > ### Author Response · Authors · 2026-04-04
> > >
> > > We sincerely appreciate the reviewer’s continued engagement and the opportunity to clarify this important point.
> > >
> > > **1. Experimental setting during the rebuttal period**
> > >
> > > Due to the limited time available during rebuttal, the additional experiments were conducted under a "1K response length, zero-RL" setting. This selection provides a clean ablation while also keeping the zero-RL setting directly comparable to our existing experiments under limited time. Admittedly, this is a relatively lightweight and challenging setting, so the absolute performance of the models is not expected to be strong. Nevertheless, we would like to emphasize that all baselines were compared under the same strictly fair and controlled setup.
> > >
> > > **2. Base Llama models indeed interact poorly with direct RL**
> > >
> > > We fully understand the reviewer’s concern regarding the relatively low absolute performance of the Llama models. However, this behavior is consistent with prior findings on the difficulty of applying direct RL to base Llama-family models. Recent literature has repeatedly shown that such models often exhibit only limited gains under zero-RL or direct RL training, and in some cases require additional mid-training to become more RL-compatible.
> > >
> > > For example, [1] shows that under a zero-RL setting, Llama 3.2-3B Base improves only from 7.4 to 10.0 on MATH500, and further argues that a dedicated mid-training stage is important for making Llama models more amenable to RL. Similarly, [2] reports only about a 1% absolute improvement under zero-RL on Llama 3.1-8B. This pattern is further corroborated by [3], which finds that direct RL on models such as Mistral-7B and Llama-3-8B often yields minimal gains and can even hurt performance. For instance, the reported gains are only 0.52% absolute on AIME24, 0.21% absolute on AIME25, and performance on GPQA Diamond even drops from 22.3% to 20.1%. Likewise, [4] reports that zero-RL on Llama 3-8B improves OlympiadBench only from 3.1% to 5.3%. A similar conclusion is drawn in [5], which notes that applying R1-Zero-like training to Llama 3-3B leads only to limited gains, such as an increase from 6.4 to 10.0 on MATH.
> > >
> > > Against this backdrop, our previous results should be interpreted under the same challenging zero-RL regime. Even in this restricted setting, our method consistently extracts more useful learning signals than standard GRPO and the other baselines, yielding up to +13.0% relative improvement on Llama 3.1-8B and +18.8% on Llama 3.2-3B.
> > >
> > >
> > > **3. Additional results in the RL with cold-start setting**
> > >
> > > To further evaluate the effectiveness of our method in the RL with cold-start setting, we use the limited rebuttal time window to conduct an additional experiment on deepseek-ai/DeepSeek-R1-Distill-Llama-8B with a 1K response length. Under this setting, we observe substantially stronger absolute performance on the Llama-based model, while the gains from our method remain consistent across benchmarks.
> > >
> > >
> > > | Method | AMC23 | MATH500 | Olympiad | Minerva | GSM8K | Avg | ΔAvg(%) |
> > > |:---|:---|:---|:---|:---|:---|:---|:---|
> > > | GRPO | 42.4 | 62.8 | 28.9 | 24.3 | 78.0 | 47.28 | - |
> > > | +random | 41.6 | 62.5 | 29.4 | 24.1 | 78.2 | 47.16 | -0.25% |
> > > | +high entropy | 42.8 | 63.2 | 30.4 | 24.6 | 78.5 | 47.90 | +1.31% |
> > > | +Coupled Rhythm Credit (Ours) | 45.6 (+3.2) | 64.2 (+1.4) | 31.9 (+3.0) | 25.8 (+1.5) | 79.2 (+1.2) | 49.34 (+2.06) | +4.36% |
> > >
> > > We hope this clarification, together with the supporting literature, helps contextualize the absolute numbers observed under the limited optimization space of the zero-RL setting. We also hope that these additional results further address the reviewer’s concern by showing that, under a stronger cold-start RL setting, the absolute performance becomes much more competitive while our method continues to deliver consistent improvements. In the revision, we will further expand this line of evaluation to provide a more complete picture. We thank the reviewer again for the thoughtful feedback, which has helped us strengthen both the paper and its empirical evaluation.
> > >
> > > [1] OctoThinker: Mid-training Incentivizes Reinforcement Learning Scaling. 2025.
> > >
> > > [2] Beyond the 80/20 Rule: High-Entropy Minority Tokens Drive Effective Reinforcement Learning for LLM Reasoning. 2025.
> > >
> > > [3] PRISM: Demystifying Retention and Interaction in Mid-Training. 2025
> > >
> > > [4] SimpleRL-Zoo: Investigating and Taming Zero Reinforcement Learning for Open Base Models in the Wild. 2025.
> > >
> > > [5] Understanding R1-Zero-Like Training: A Critical Perspective. 2025.

---

### Decision · Program_Chairs · 2026-04-30

**Decision:**

Accept (regular)

**Comment:**

This paper tackles the interesting problem of improving model reasoning abilities through understanding models' attention mechanisms. The reviewing team recognized the authors' significant effort and found clear value in their clear presentation, simple and intuitive method, good experimental results, and interesting analysis findings. The reviewers appreciated the authors' thoughtful response, which successfully addressed some primary concerns. Some reviewers hope to see results on more model families, while I find two families to be adequate for showing model effectiveness. A reviewer only challenges that attention is not a solid way for XAI. While I agree with the reviewer that attention maps do not show causal relations, I find it acceptable that the understanding of the model is derived based on analyzing attention dynamics.